# *Hox*-dependent coordination of mouse cardiac progenitor cell patterning and differentiation

Sonia Stefanovic[1†], Brigitte Laforest[1†‡], Jean-Pierre Desvignes[1], Fabienne Lescroart[1], Laurent Argiro[1], Corinne Maurel-Zaffran[2], David Salgado[1], Elise Plaindoux[1], Christopher De Bono[2], Kristijan Pazur[3], Magali Théveniau-Ruissy[1,2], Christophe Béroud[1], Michel Puceat[1], Anthony Gavalas[3], Robert G Kelly[2], Stephane Zaffran[1*]

[1]Aix Marseille Univ, INSERM, Marseille Medical Genetics, Marseille, France; [2]Aix Marseille Univ, CNRS UMR7288, IBDM, Marseille, France; [3]Paul Langerhans Institute Dresden (PLID) of Helmholtz Center Munich at the University Clinic Carl Gustave Carus of TU Dresden, Helmoholtz Zentrum München, German Center for Diabetes Research (DZD), Dresden, Germany

**Abstract** Perturbation of addition of second heart field (SHF) cardiac progenitor cells to the poles of the heart tube results in congenital heart defects (CHD). The transcriptional programs and upstream regulatory events operating in different subpopulations of the SHF remain unclear. Here, we profile the transcriptome and chromatin accessibility of anterior and posterior SHF sub-populations at genome-wide levels and demonstrate that Hoxb1 negatively regulates differentiation in the posterior SHF. Spatial mis-expression of *Hoxb1* in the anterior SHF results in hypoplastic right ventricle. Activation of *Hoxb1* in embryonic stem cells arrests cardiac differentiation, whereas *Hoxb1*-deficient mouse embryos display premature cardiac differentiation. Moreover, ectopic differentiation in the posterior SHF of embryos lacking both *Hoxb1* and its paralog *Hoxa1* results in atrioventricular septal defects. Our results show that Hoxb1 plays a key role in patterning cardiac progenitor cells that contribute to both cardiac poles and provide new insights into the pathogenesis of CHD.

**\*For correspondence:**
stephane.zaffran@univ-amu.fr

[†]These authors contributed equally to this work

**Present address:**
[‡]Cardiometabolism Group, Internal Medicine Research Unit, Pfizer Research and Development, Cambridge, United States

## Introduction

Heart morphogenesis and patterning require precise temporal differentiation of distinct cardiac progenitor populations that arise from two early sources of mesoderm progenitors, the first heart field (FHF) and the second heart field (SHF) (*Buckingham et al., 2005*). The FHF originates from the anterior splanchnic mesoderm and forms the cardiac crescent. The SHF is a progenitor population originating in pharyngeal mesoderm that contributes to heart tube elongation through the progressive addition of cells from the dorsal pericardial wall to both poles of the forming heart. The SHF gives rise to right ventricular and outflow tract myocardium at the arterial pole, and to atrial myocardium including the dorsal mesenchymal protrusion (DMP) at the venous pole (*Zaffran and Kelly, 2012*). In the absence of SHF cell addition, impaired heart tube elongation and looping leads to early embryonic lethality, and perturbation of this process underlies a spectrum of common congenital heart defects (CHDs) (*Prall et al., 2007*; *Cai et al., 2003*). Lineage tracing analysis in mammals has revealed that the SHF is sub-divided into distinct anterior and posterior regions (aSHF and pSHF) (*Domínguez et al., 2012*; *Lescroart et al., 2012*; *Vincent and Buckingham, 2010*). Cardiac progenitors in the aSHF contribute to the right ventricular and outflow tract myocardium (*Buckingham et al., 2005*), whereas pSHF cells participate to the formation of the atrial and

atrioventricular septation through development of the DMP that forms the muscular base of the primary atrial septum (*Briggs et al., 2012*). In addition, a sub-population of the pSHF has been shown to contribute to the inferior wall of the outflow tract which gives rise to the myocardium at the base of the pulmonary trunk (*Bertrand et al., 2011*).

A complex network of signaling inputs and transcriptional regulators is required to regulate SHF development (*Rochais et al., 2009*). Among these signaling molecules, retinoic acid (RA) has been shown to pattern the SHF (*Stefanovic and Zaffran, 2017*; *Hochgreb et al., 2003*). Specifically, RA signaling is required to define the limit between the anterior and posterior SHF, as indicated by the abnormal posterior expansion of the expression of aSHF markers genes, including *Fgf8*, *Fgf10*, and *Tbx1*, in *Raldh2*-mutant embryos (*De Bono et al., 2018*; *Ryckebusch et al., 2008*; *Sirbu et al., 2008*). *Hoxa1*, *Hoxa3* and *Hoxb1* are expressed in overlapping sub-populations of cardiac progenitor cells in the pSHF and downregulated prior to differentiation (*Bertrand et al., 2011*). *Hoxb1*- and *Hoxa1*-expressing progenitor cells located in the pSHF segregate to both cardiac poles, contributing to the inflow tract and the inferior wall of the outflow tract (*Lescroart and Zaffran, 2018*; *Bertrand et al., 2011*). In contrast, cardiac progenitors that contribute to the superior wall of the outflow tract and right ventricle do not express Hox transcription factors. *Hoxb1* is required for normal deployment of SHF cells during outflow tract development (*Roux et al., 2015*). TALE-superclass transcription factors (three-amino acid length extension) such as Pbx1-3 or Meis1-2, which are co-factors of anterior Hox proteins, are also expressed in cardiac progenitors, suggesting a wider role for HOX/TALE complexes during SHF development (*Paige et al., 2012*; *Wamstad et al., 2012*; *Stankunas et al., 2008*).

Identification of SHF-restricted regulatory elements has provided evidence that different transcriptional programs operate in distinct SHF sub-populations. Cells expressing *Cre* recombinase under the control of a SHF-restricted regulatory element from the *Mef2c* gene contribute widely to the outflow tract and right ventricle, as well as to a population of cells at the venous pole of the heart giving rise to the primary atrial septum and DMP (*De Bono et al., 2018*; *Goddeeris et al., 2008*; *Verzi et al., 2005*; *Dodou et al., 2004*). Although subdomains of the SHF prefigure and are essential to establish distinct structures within the mature heart, it is unclear how distinct sub-populations are defined. Here, we identify the genome-wide transcriptional profiles and chromatin accessibility maps of sub-populations of SHF cardiac progenitor cells using RNA- and ATAC-sequencing approaches on purified cells. Through gain and loss of function experiments we identify Hoxb1 as a key upstream player in SHF patterning and deployment. Mis-expression of *Hoxb1* in the Hox-free domain of the SHF results in aberrant cellular identity of progenitor cells and arrested cardiac differentiation, leading ultimately to cell death. The addition of progenitor cells from the pSHF to the venous pole is also impaired in *Hoxa1*$^{-/-}$; *Hoxb1*$^{-/-}$ hearts, resulting in abnormal development of the DMP and consequent atrioventricular septal defects (AVSDs). Hoxb1 is thus a critical determinant of cardiac progenitor cell fate in vertebrates.

## Results

### Transcriptomic and epigenetic profiling of the SHF

To identify the transcriptional profiles of distinct cardiac progenitor populations, we made use of two transgenic mouse lines, *Hoxb1*$^{GFP}$ and *Mef2c-AHF-Cre* (*Mef2c-Cre*), that drive reporter gene expression in sub-domains of the SHF (*Roux et al., 2015*; *Bertrand et al., 2011*; *Briggs et al., 2013*; *Verzi et al., 2005*). At embryonic day (E) 9.5 (16 somites [s]), the GFP reporter of *Hoxb1*$^{GFP}$ embryos is detectable in the posterior region of the SHF (*Figure 1A*). Genetic lineage analysis of *Hoxb1*-expressing cells using the *Hoxb1*$^{IRES-Cre}$ mouse line showed that *Hoxb1* progenitors contribute to both atria, the DMP and the myocardium at the base of the pulmonary trunk at E11.5-E12.5 (*Figure 1B,C*). Genetic lineage analysis of *Mef2c-Cre*-labelled cells using *Mef2c-Cre;Rosa*$^{tdT}$ mouse line showed that Tomato-positive (Tomato+) cells are detected in the arterial pole of the heart and the DMP at E9.5-E10.5 (*Figure 1D,E*). At E12.5, the contribution of *Mef2-Cre*-expressing cells is observed in the great arteries (aorta and pulmonary trunk) and the right ventricle (*Figure 1F*), consistent with previous observations (*De Bono et al., 2018*; *Goddeeris et al., 2008*; *Verzi et al., 2005*). To further characterize the expression pattern of these two reporter lines we performed RNA-FISH (RNAscope fluorescent in situ hybridization). At E8.5–9, RNA-FISH showed that expression of *Osr1*,

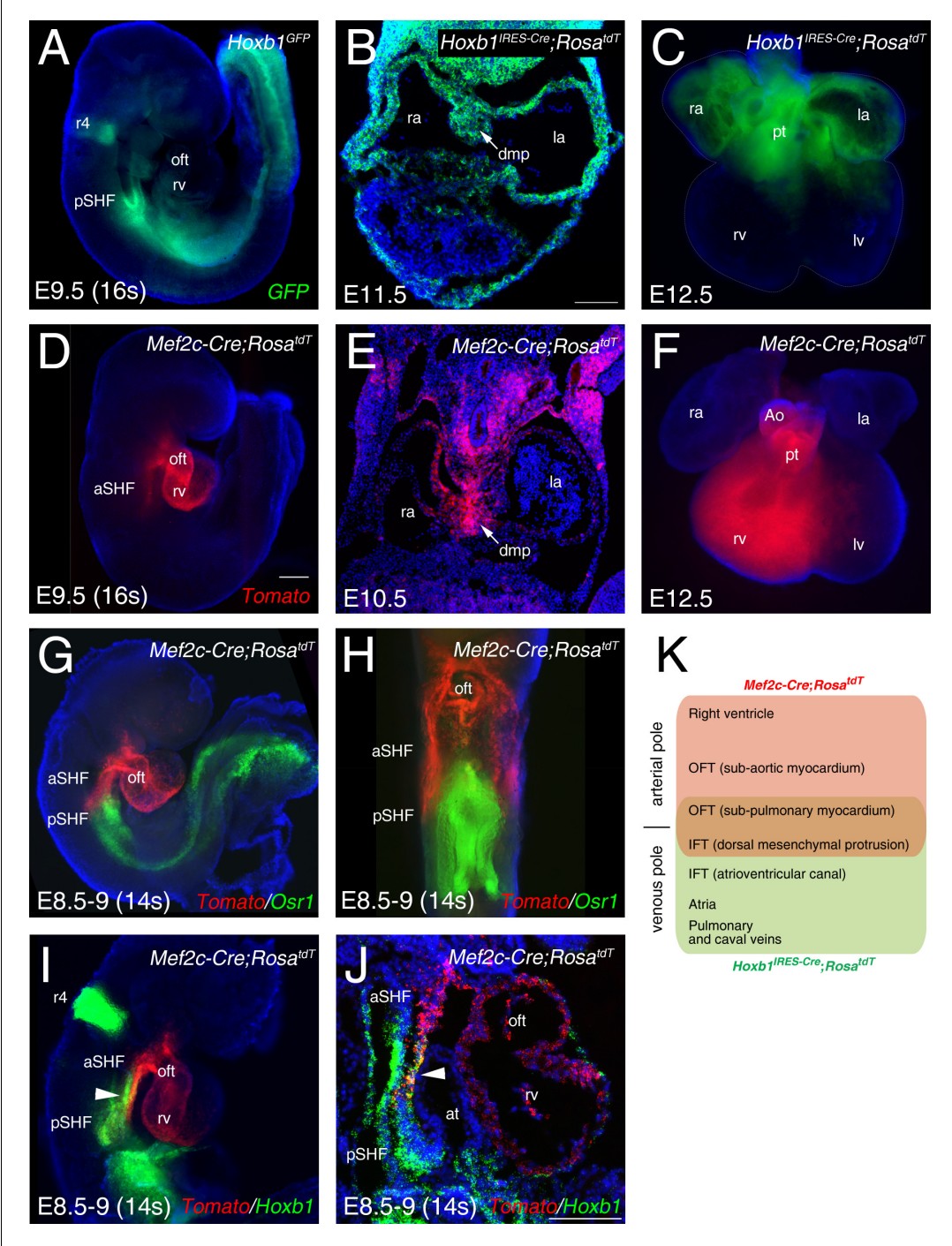

**Figure 1.** Characterization of two transgenic lines defining complementary domains of the SHF. (**A**) Whole-mount fluorescence microscopy of E9.5 (16 somites [s]) *Hoxb1^GFP* embryos. (**B**) Transverse section at E11.5 heart showing *Hoxb1-Cre* genetic lineage contribution to atrial myocardium and the dorsal mesenchymal protrusion (DMP). (**C**) Ventral view of an E12.5 heart showing the *Hoxb1-Cre* (*Hoxb1^IRES-Cre*;*Rosa^tdT* - green) genetic lineage contributions to both atria and sub-pulmonary myocardium. (**D**) E9.5 (16s) *Mef2c-Cre;Rosa^tdT* embryos showing the contribution of the *Mef2c-Cre* genetic lineage (*Tomato*, red) to the outflow tract and the right ventricle. (**E**) Transverse section at E10.5 showing the *Mef2c-Cre* genetic lineage contribution to the DMP. (**F**) Ventral view of an E12.5 heart showing the *Mef2c-Cre* genetic lineage contribution to the right ventricle and great arteries. (**G**) RNA-FISH showing the expression of *Osr1* (green) and the *Mef2c-Cre* labeled cells (*Tomato*; red). (**H**) Ventral view of the embryo shown in G. Distribution of *Osr1* is detected in the posterior domain of *Mef2c-Cre*. (**I,J**) RNA-FISH showing a small domain of overlap between *Hoxb1* (green) and *Mef2c-Cre* labeled cells (Tomato; red). (**K**) Cartoon summarizing the contribution of the *Hoxb1-Cre* (green) and *Mef2c-Cre* (red) lineages in the embryo at E9.5. Nuclei are stained with Hoechst. ao, aorta; at, atria; aSHF, anterior second heart field; ift, inflow tract; la, left atria; lv, left ventricle; oft, outflow

*Figure 1 continued on next page*

Figure 1 continued

tract; pt, pulmonary trunk; pSHF, posterior second heart field; r4, rhombomere 4; ra, right atria; rv, right ventricle; Scale bars represent 100 µm (C, J); 200 µm (D).

a gene expressed in pSHF progenitors (*Zhou et al., 2015*), largely overlapped with *Hoxb1* expression (*Figure 1G and I*), whereas *Mef2c-Cre* predominantly labeled a distinct progenitor cell population to *Osr1* (*Figure 1G,H*). Double whole-mount in situ hybridization identified a minor subset of cardiac progenitors co-labeled by *Hoxb1* and *Mef2c-Cre;Rosa^{tdT}* (*Tomato+*), likely corresponding to progenitor cells giving rise the DMP at the venous pole and inferior outflow tract wall at the arterial pole (*Figure 1I–K*; *Roux et al., 2015*; *Briggs et al., 2013*; *Bertrand et al., 2011*; *Verzi et al., 2005*).

After micro-dissection and dissociation of the SHF region from *Hoxb1^{GFP}* (GFP+) and *Mef2c-Cre; Rosa^{tdT}* (Tomato+) embryos at E9.5 (16s; n = 3 each), GFP+ and Tomato+ cells were purified by flow cytometry-activated cell sorting (FACS) and subsequently used for RNA-seq (*Figure 2A*). FACS analysis showed that Tomato+ and GFP+ cells comprise, respectively, 33% and 23% of the total micro-dissected region (*Figure 2B,D*). The enrichment was validated by qPCR for a set of genes known to be specifically expressed in the pSHF (*Hoxa1, Hoxb1, Osr1, Tbx5* and *Aldh1a2*) (*Figure 2C,E*). Although the pSHF markers *Tbx5* and *Osr1* were not detected in the Tomato+ cells, *Hoxa1, Hoxb1* and *Aldh1a2* transcripts were amplified in cells from that micro-dissected region (*Figure 2C*). Detection of *Hoxb1* mRNAs in the Tomato+ cells confirms the co-labeling observed between *Hoxb1* and *Mef2c-Cre;Rosa^{tdT}* (*Tomato+*) expression in a subset of cardiac progenitors that contribute to both poles of the heart tube (*Figure 1I–K*). Principal component analysis (PCA) and calculation of the Euclidean distance between the regularized log (rlog)-transformed data for each sample using DESeq2 demonstrated the strong similarity between biological replicates (*Figure 2—figure supplement 1A,B*). We identified 11,970 genes expressed in both cell types. 2,133 genes were transcribed specifically in the GFP+ population (*Figure 2F* and *Figure 2—figure supplement 1C*). Gene ontology (GO) enrichment analysis for the biological processes linked to these genes showed a significant enrichment of GO terms associated with 'heart development', 'epithelium development', 'cardiac chamber morphogenesis' and 'cell adhesion' (*Figure 2G*). Included in the 'heart development' list we identified several genes previously described as being expressed in the posterior region of the SHF (*e.g., Tbx5, Osr1, Tbx18, Foxf1,* and *Wnt2*), as well as *Bmp4, Nr2f2, Sema3c, Gata4* (*Figure 2H*- and *Figure 2—figure supplement 2*; *Supplementary file 1*). RNA-FISH analysis validated the expression of *Bmp4* in the pSHF (*Figure 2—figure supplement 2*).

Given the small overlap between *Hoxb1* and Tomato expression (*Figure 1J*), we generated triple transgenic *Hoxb1^{GFP};Mef2c-Cre;Rosa^{tdT}* embryos at stage E9.5 (*Figure 2I*) and purified double positive (GFP+/Tomato+) and simple positive (GFP+ or Tomato+) cells. FACS analysis showed that the GFP+/Tomato+ gate comprised only 1% of the total micro-dissected region (*Figure 2J*). Interestingly, transcriptional analysis revealed that both *Hoxb1* and *GFP* transcripts were decreased in the GFP+/Tomato+ population compared to the GFP+ population. Conversely, *Cre* transcripts were equally expressed in both the Tomato+/GFP+ and Tomato+ populations suggesting that the *Mef2c-AHF* enhancer was still active in pSHF cells at this stage (*De Bono et al., 2018*; *Rana et al., 2014*; *Briggs et al., 2012*; *Xie et al., 2012*; *Goddeeris et al., 2008*).

To define accessible sites for transcriptional regulation in SHF sub-populations, we performed ATAC-seq (*Buenrostro et al., 2015*). We performed ATAC-seq on FACS-sorted Tomato+ and GFP+ cells from E9.5 (12–14 s) embryos. Samples were subjected to massively parallel sequencing and overlapping peaks from replicate samples were merged to identify high-confidence regions of open chromatin. The correlation heatmaps and PCA plot highlighted the differences in ATAC read concentrations between Tomato+ and GFP+ samples (*Figure 3—figure supplement 1A,B*). ATAC-seq peaks representing open chromatin were highly reproducible between biological replicates and showed a clear enrichment at putative regulatory elements (*Figure 3—figure supplement 1C*). We performed a stringent analysis to identify quantitative differences in chromatin accessibility. By comparing the signal for each peak in Tomato+ and GFP+ populations, we identified 1285 peaks that were exclusively accessible in the GFP+ population (*Figure 3A,B*). Approximately 94% of peaks found in both the GFP+ and Tomato+ populations, while 3.5% of the peaks were exclusively present in the GFP+ population (*Figure 3B*). We then asked whether DNA regions differentially accessible

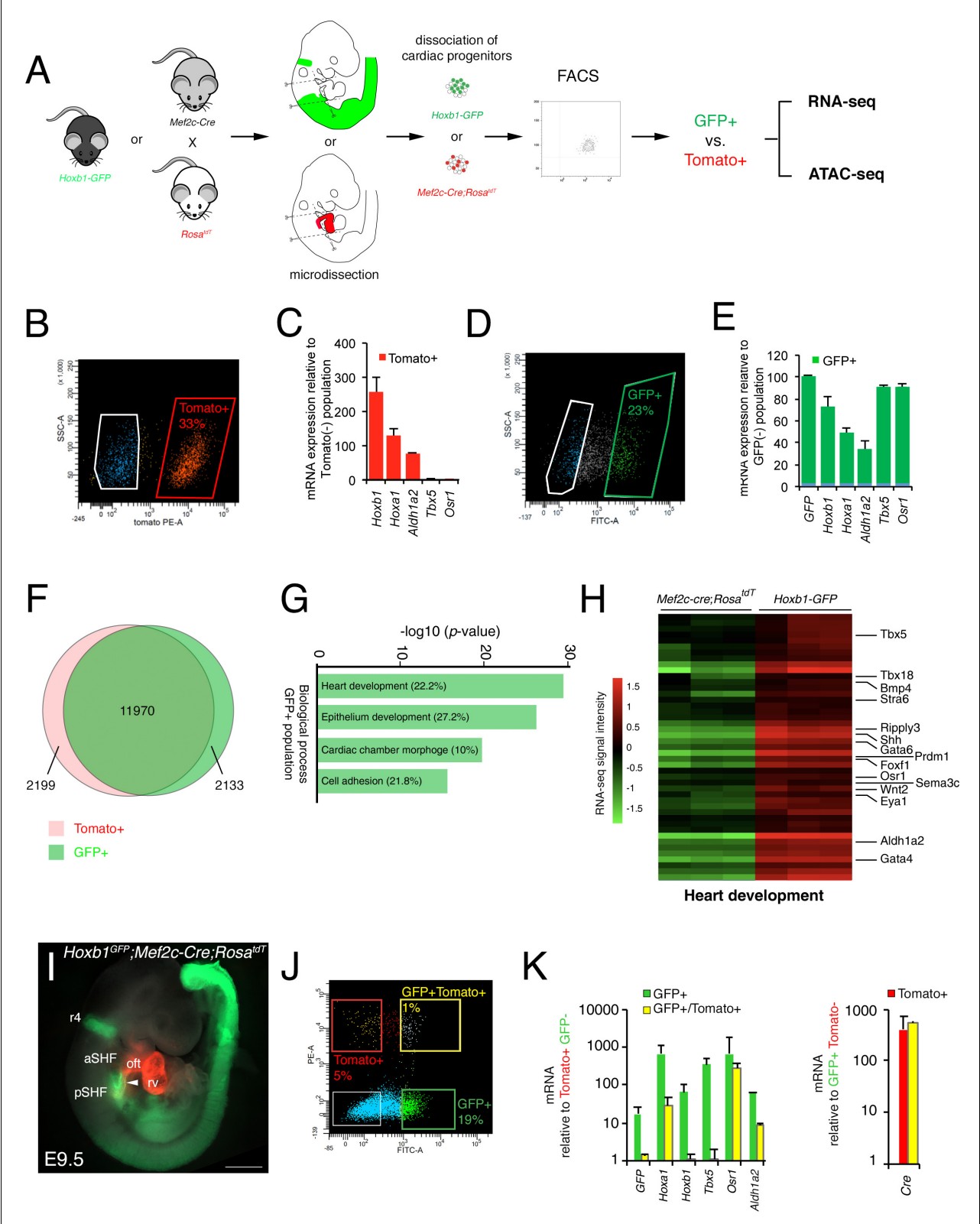

**Figure 2.** Molecular signature of the posterior SHF. (**A**) Scheme of the protocol utilized to characterize the molecular signature of the SHF on isolated *Mef2c-Cre;Rosa^tdT* (Tomato) and *Hoxb1^GFP* (GFP) positive cells. (**B,D**) FACS profile of E9.5 cardiac progenitor cells isolated from *Mef2c-Cre;Rosa^tdT* and *Hoxb1^GFP* embryos. (**C,E**) Expression of pSHF markers (*Hoxa1, Hoxb1, Osr1, Aldh1a2, Tbx5, GFP*) was analyzed by real-time qPCR. Data were normalized to *HPRT* and expressed as folds of increase over negative population. (**F**) Venn diagram showing transcripts differentially expressed in the *Figure 2 continued on next page*

Figure 2 continued

GFP+ (green) compared to the Tomato+ (red) populations. (G) Gene ontology (GO) analysis of GFP+ progenitor cells performed with ClusterProfiler system showing enrichment of upregulated genes in the pSHF with ranked by -log$_{10}$ (p-value). The percentage corresponds to the 'BG ratio' obtained in the GO analysis. (H) Example of the heatmap of 'heart development' GO term associated genes analyzed by RNA-seq (n = 3 from GFP+ cells, n = 3 from Tomato+ cells). (I) Whole-mount fluorescence microscopy of triple transgenic Hoxb1$^{GFP}$;Mef2c-Cre;Rosa$^{tdT}$ embryos at stage E9.5. (J) FACS profile of E9.5 cardiac progenitor cells isolated from Hoxb1$^{GFP}$;Mef2c-Cre;Rosa$^{tdT}$ embryos. (K) Expression of pSHF markers (Hoxa1, Hoxb1, Osr1, Aldh1a2, Tbx5), GFP and the Cre recombinase were analyzed with real-time qPCR. Data were normalized to HPRT and expressed as folds of increase over negative population. aSHF, anterior SHF; oft, outflow tract; pSHF, posterior SHF; rv, right ventricle. Scale bars: 500 μm.

The online version of this article includes the following figure supplement(s) for figure 2:

**Figure supplement 1.** Quality assessment of RNA-seq data performed with purified cardiac progenitor cells.

**Figure supplement 2.** Spatial validation of marker gene expression in cardiac progenitor populations.

between the Tomato+ and GFP+ populations were selectively associated with genes specifically expressed in the pSHF that were identified from our RNA-seq analysis. In order to assess differential chromatin accessibility at each consensus peaks we used an affinity analysis as a quantitative approach (*Figure 3C*; see Materials and methods). Quantitative analysis confirmed the number of peaks identified by the qualitative approach for each GFP+ or Tomato+ population. In addition, quantitative differences in peak signals were observed between the Tomato+ and GFP+ populations (*Figure 3—figure supplement 1D*). We next asked if global differences between the accessible chromatin landscapes of the Tomato+ and GFP+ populations correlated with changes in gene expression. Thus, we mapped individual ATAC-seq peaks in the Tomato+ and GFP+ populations based on their distance to the nearest transcription start site (TSS) and examined the expression of the corresponding genes (*Figure 3D,E*). Changes in chromatin accessibility did not correlate precisely with changes in gene expression and several peaks near differentially expressed genes were not differentially accessible. Such decoupling between enhancer accessibility and activity has been observed in other developmental contexts, including early cardiogenesis (*Racioppi et al., 2019*; *Paige et al., 2012*; *Wamstad et al., 2012*). However, we found 53 (Tomato+ population) and 65 (GFP+ population) peaks correlating with changes in gene expression (*Figure 3E*). Interestingly, we observed an over-representation of TSS in the 'GFP+' peaks compared to the 'Tomato+' peaks suggesting that genes with open promoters are preferentially activated in the pSHF (*Figure 3—figure supplement 1E*). Among the 65 peaks specific to the GFP+ population we found Hoxb1, Aldh1a2 and Sema3c, loci, which showed open chromatin regions concentrated in the promoter and regulatory regions occupied by several transcription factors (*Figure 3D–F*; 3H). ATAC-seq data for the GFP+ population thus revealed a high read count around the promoter regions of genes enriched in the pSHF, including Hoxb1, Aldh1a2, Osr1 and Sema3c (*Figure 3F–H*). Similarly, the pSHF enhancer previously identified at the Foxf1a locus (*Hoffmann et al., 2014*) exhibited enrichment of ATAC-seq reads in GFP+ population (*Figure 3—figure supplement 1F*; *Supplementary file 2*). In contrast, the established *Mef2c* anterior heart field enhancer (Mef2c-F6; 285 bp) (*Dodou et al., 2004*) was marked by open chromatin in the Tomato+ population, but not the GFP+ population (*Figure 3—figure supplement 1F*), confirming that ATAC-seq predominantly marks active promoters and enhancers in prominent compartment-specific patterns. Among the ATAC-seq peaks we observed a high read density within an intron of the *Mef2c* gene, approximately 2.5 kb or 4 kb upstream of the previously described *Mef2c*-AHF enhancers (*Dodou et al., 2004*; *von Both et al., 2004*).

Together this analysis indicates that our dataset can be used to identify regulatory elements in distinct SHF sub-populations. ATAC-seq in GFP+ cells identified several pSHF-specific peaks indicating that these regions may function as enhancers (*Figure 3F–H*). We then used HOMER to identify enriched putative transcription factor motifs in the open chromatin regions of the GFP+ and Tomato + populations (*Figure 3I*; *Figure 3—figure supplement 1G*). One of the most highly enriched motif in open chromatin regions of the GFP+ population was the consensus HOX motif (*Figure 3I*; *Fan et al., 2012*). Other significantly enriched motifs include putative binding sites for PBX and MEIS proteins (*Figure 3I*), TALE-class transcription factors interacting with HOX proteins (*Lescroart and Zaffran, 2018*; *Ladam and Sagerström, 2014*). Members of the PBX and MEIS families have been previously identified as cofactors of Hoxb1 in mammalian cell lines and embryonic tissues, indicating a high level of functional conservation (*Roux and Zaffran, 2016*; *Ladam and*

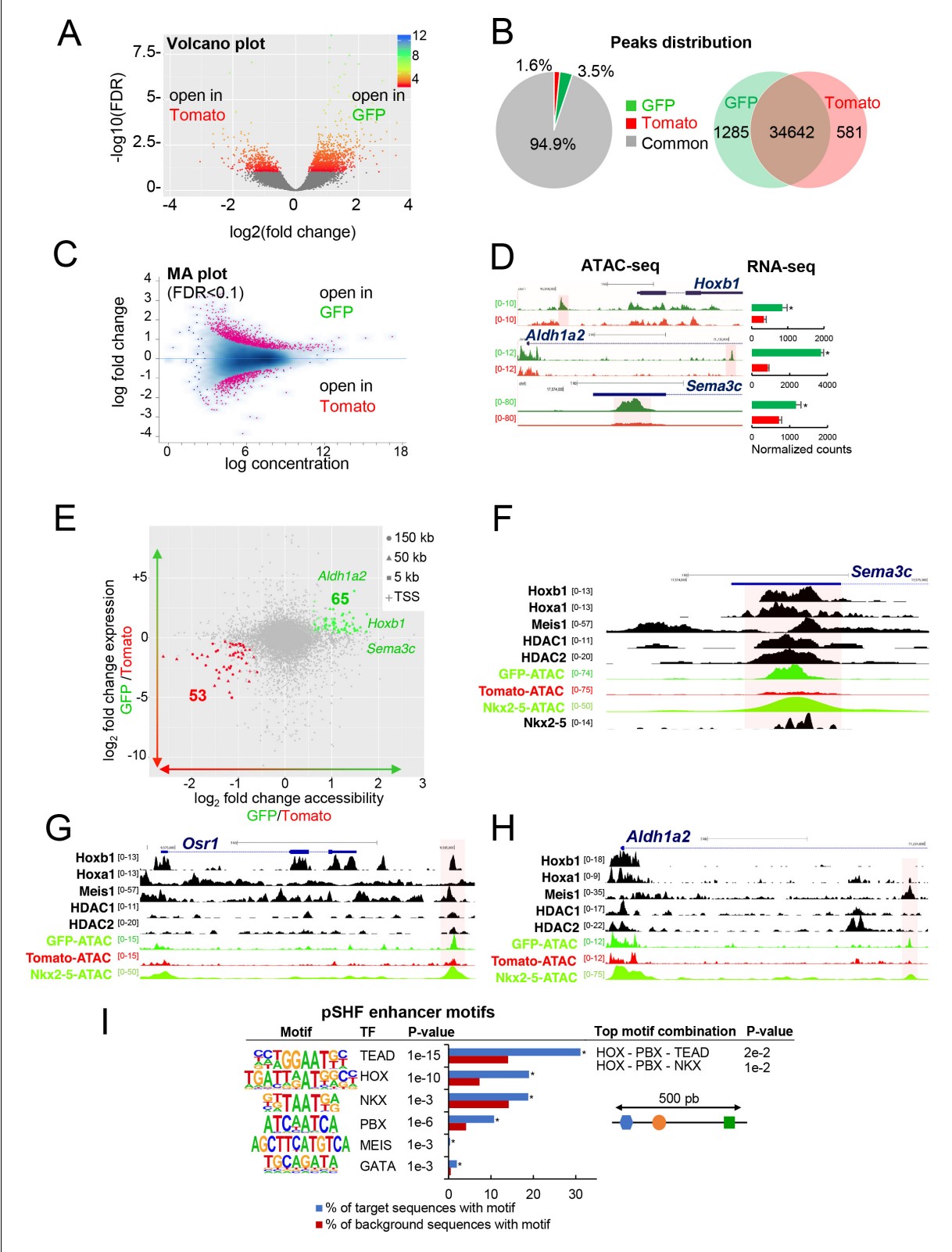

**Figure 3.** Differential chromatin accessibility in GFP+ and Tomato+ cardiac progenitor cells. (**A**) Volcano plot of ATAC-seq performed from GFP+ and Tomato+ cardiac progenitors. (**B**) Pie chart showing the distribution of the ATAC-seq peaks in the two populations. Venn diagram showing overlap of the ATAC-seq peaks in the two populations. (**C**) MA plot of ATAC-seq peaks in GFP+ *versus* Tomato+ cells. (**D**) Open chromatin profiles correlate with transcriptional expression. Browser views of *Hoxb1*, *Aldh1a2* and *Sema3c* loci with ATAC-seq profiles of GFP+ pSHF progenitor cells (green) and

*Figure 3 continued on next page*

Figure 3 continued

Tomato+ *Mef2c-Cre* labeled cells (red). Data represent merged technical and biological replicates for each cell type. The *y*-axis scales range from 0 to 80 in normalized arbitrary units. The tracks represent ATAC-seq, whereas the bar graphs represent RNA-seq. Boxed regions show cell-type-specific peaks around *Aldh1a2*, *Osr1*, and *Sema3C* gene loci. (E) Change in accessibility versus change in gene expression in GFP+ and Tomato+ cells. For each ATAC-seq peak, the log of the ratio of normalized ATAC-seq reads (GFP/Tomato) is plotted on the x-axis, and the log of the ratio normalized RNA-seq reads corresponding to the nearest gene is plotted in the y-axis. Peaks that are both significantly differentially accessible (FDR < 0.1) and significantly differentially expressed (FDR < 0.1) are colored green (more open in GFP+ cells, higher expression in GFP+ cells; 65 peaks) or red (more open in Tomato population, higher expression in Tomato; 53 peaks). (F–H) Browser views of *Sema3c* (F), *Osr1* (G) and *Aldh1a2* (H) gene loci with ATAC-seq profiles of GFP+ pSHF progenitor cells (green) and Tomato+ *Mef2c-Cre* labeled cells (red). Open chromatin profiles correlate with transcription factor binding at putative enhancers specific for cardiac progenitor cells. (I) pSHF enhancers were enriched in DNA binding motifs for HOX and known cardiac transcription factors such as PBX and MEIS. HOX recognition motifs were strongly enriched in a known motif search in pSHF enhancers. Other enriched matrices match binding sites of known cardiac regulators. HOX binding motifs are highly enriched at genomic regions bound by cardiac transcription factors. *p*-values were obtained using HOMER (*Heinz et al., 2010*). Combinations of 3 sequence motifs contained within 500 bp are shown.

The online version of this article includes the following figure supplement(s) for figure 3:

**Figure supplement 1.** Quality assessment of ATAC-seq data performed with purified cardiac progenitor cells.

*Sagerström, 2014*; *Mann et al., 2009*). In addition, we found overrepresentation of motifs for GATA and TEA domain (TEAD) family transcription factors as well as for the NKX2-5 homeodomain. Because transcription factors function in a combinatorial manner, we identified combinations of multiple motifs that were most enriched at pSHF candidate enhancers relative to non-pSHF enhancers (*Figure 3I*). Our computational analysis showed that the most enriched combinations contained HOX motifs adjacent to TALE transcription factor recognition sequences. Consistent with these observations, chromatin immunoprecipitation (ChIP)-sequence data for the cofactors of HOX proteins (Meis1, Nkx2-5, HDACs) revealed that these factors bind putative regulatory elements marked by open chromatin in the GFP+ but not in the Tomato+ population (*Figure 3F–H*). In open chromatin regions of the Tomato+ population we observed enrichment of motifs for RUNX, LHX and TEAD transcription factors (*Figure 3—figure supplement 1G*). However, we did not identify combinations of multiple motifs that were enriched in these regions relative to non-specific enhancers.

## Mis-expression of *Hoxb1* in the *Mef2c-AHF-Cre* lineage disrupts right ventricular formation

In order to investigate the role of Hoxb1 during heart development we generated a conditional activated *Tg(CAG-Hoxb1-EGFP)^{1Sza}* (*Hoxb1^{GoF}*) transgenic mouse, in which *Hoxb1* cDNA expression is activated upon Cre-mediated recombination. As previously reported *Hoxb1^{GoF}* mice without any Cre allele were viable and healthy (*Zaffran et al., 2018*). *Hoxb1^{GoF}* mice were crossed with to *Mef2c-AHF-Cre* (*Mef2c-Cre*) mice to mis-express *Hoxb1* in *Mef2c-AHF*+ cells (*Verzi et al., 2005*). *Hoxb1^{GoF}*;*Mef2c-Cre* embryos exhibited severe heart defects as early as E9.5, as observed by a looping defect and common ventricular chamber in transgenic compared to control embryos (*Figure 4A–B*). Expression of GFP in the aSHF and its derivatives confirmed *Mef2c-Cre*-driven recombination (*Figure 4A',B'*). Immunostaining of E10.5 control embryos revealed normal future right and left ventricular chambers with developing trabeculae (*Figure 4C*). However, in *Hoxb1^{GoF}*; *Mef2c-Cre* embryos, the heart was abnormally shaped with no clear distinction between right and left ventricular chambers (*Figure 4D*). The phenotype was even more pronounced at E12.5, when Hematoxylin and Eosin-stained transverse sections showed a hypoplastic right ventricle with abnormal positioning of the ventricular septum (*Figure 4E and F*). Hypoplasia of the right ventricle in *Hoxb1^{GoF}*;*Mef2c-Cre* embryos was evident at E15.5 (*Figure 4G–J*; 8/8). At this stage, an abnormally thin right ventricular wall was observed (5/8). In addition, 50% of *Hoxb1^{GoF}*;*Mef2c-Cre* embryos showed misalignment of the great arteries (4/8) and 63% displayed ventricular septal defects (VSD; 5/8). In this context, we never observed viable *Hoxb1^{GoF}*;*Mef2c-Cre* pups with hypoplastic right ventricle. Overall, these results suggest that ectopic *Hoxb1* expression in the *Mef2c-AHF* lineage disrupts the contribution of anterior cardiac progenitor cells to the forming heart.

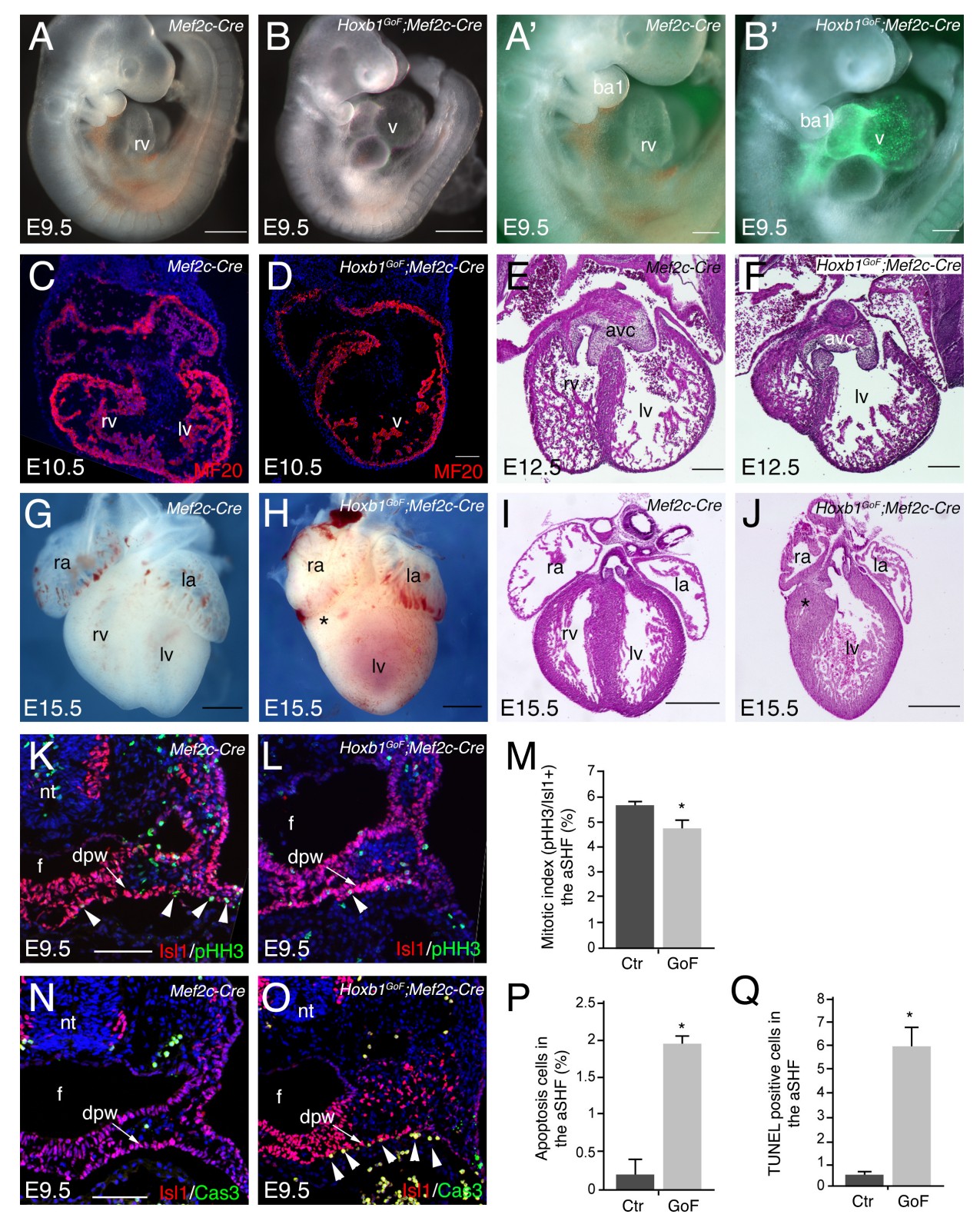

**Figure 4.** Activation of *Hoxb1* expression in aSHF progenitors disrupts the formation of the right ventricle. (**A,B**) Macroscopic view of *Mef2c-AHF-Cre* (*Mef2c-Cre*, control) and *Hoxb1^GoF^;Mef2c-Cre* embryos at E9.5. (**A',B'**) High magnification of embryo in A and B showing GFP activity in the Cre-recombinase driven cells. (**C,D**) Immunofluorescence with MF20 (red) in control (**C**) and *Hoxb1^GoF^;Mef2c-Cre* (**D**) hearts at E10.5. (**E–J**) Hematoxylin and Eosin (H and E) staining on transversal section of control (**E,I**) and *Hoxb1^GoF^;Mef2c-Cre* (**F,J**) hearts at E12.5 and E15.5. The right ventricle (asterisk) in
*Figure 4 continued on next page*

Figure 4 continued

Hoxb1$^{GoF}$;Mef2c-Cre hearts is hypoplastic compared to control hearts (n = 8). (G,H) Whole-mount views of control (G) and Hoxb1$^{GoF}$;Mef2c-Cre (H) hearts at E15.5. (K,L) Immunofluorescence with Isl1 (red) and Phospho-histone H3 (pHH3, green) on Mef2c-Cre (K) and Hoxb1$^{GoF}$;Mef2c-Cre (L) embryos at E9.5. (M) Quantification of pHH3-positive cells in the aSHF Isl1+, showed a reduced of the mitotic index in Hoxb1$^{GoF}$;Mef2c-Cre (n = 3; GoF) compared to Mef2c-Cre (Ctr) (n = 7) embryos. (N,O) Immunofluorescence with Isl1 (red) and Caspase 3 (Cas3, green) on Mef2c-Cre (N) and Hoxb1$^{GoF}$; Mef2c-Cre (O) embryos at E9.5. Arrowheads indicate Cas3-positive cells. (P) Quantification of Cas3-positive cells revealed increased cell death in the aSHF of Hoxb1$^{GoF}$;Mef2c-Cre embryos. (Q) Quantification of TUNEL staining performed on Mef2c-Cre (Ctr) and Hoxb1$^{GoF}$;Mef2c-Cre embryos (GoF). ao, aorta; avc, atrioventricular canal; ba, branchial arch; dpw, dorsal pericardial wall; f, foregut pocket; la, left atrium; lv, left ventricle; nt, neural tube; oft, outflow tract; PM, pharyngeal mesoderm; pt, pulmonary trunk; ra, right atrium; rv, right ventricle. Scale bars: 100 µm (A',B',D,E,F); 200 µm (A',B',E,F); 500 µm (A,B,G-J).

The online version of this article includes the following figure supplement(s) for figure 4:

Figure supplement 1. Reduction of SHF length in Hoxb1$^{GoF}$;Mef2c-Cre embryos.

Figure supplement 2. Over-expression of Hoxb1 in the pSHF does not affect the morphology of the embryonic heart.

## Ectopic *Hoxb1* activity affects the survival of SHF progenitor cells

To study the cause of the right ventricular hypoplasia defect in Hoxb1$^{GoF}$;Mef2c-Cre embryos, we performed a lineage analysis of aSHF progenitors using the Rosa26R-lacZ (R26R) reporter line. By E9.5, β-galactosidase (β-gal)-positive cells were detected in the SHF, the outflow tract and right ventricle of control embryos, reflecting the normal contribution of cardiac progenitor cells (*Figure 4— figure supplement 1A*). In contrast, a striking reduction of the β-gal+ expression domain was observed in Hoxb1$^{GoF}$;Mef2c-Cre;R26R embryos (*Figure 4—figure supplement 1B*). The reduction was evident within the heart and in the SHF of these embryos (*Figure 4—figure supplement 1B*; asterisk). This observation was further confirmed using the Mlc1v-nlacZ-24 (Mlc1v24) transgenic line, containing an Fgf10 enhancer trap transgene expressed in the aSHF (*Figure 4—figure supplement 1C–F*). Quantitative analysis demonstrated a reduction of the β-gal+ expression domain in Hoxb1$^{GoF}$;Mef2c-Cre;Mlc1v24 compared to control embryos consistent with a reduction of the distance between the arterial and venous poles (*Figure 4—figure supplement 1G–J*). These data reveal that cardiac defects in Hoxb1$^{GoF}$;Mef2c-Cre embryos are associated with a decrease in progenitor cell numbers in the aSHF.

To determine the origin of this decrease we performed proliferation and cell death assays at E9.5. More specifically, we determined the mitotic index (pHH3+ cells) of Isl1+ cardiac progenitor cells. There was a modest reduction in cell proliferation between control and Hoxb1$^{GoF}$;Mef2c-Cre embryos (*Figure 4K–M*). However, a significant increase of Caspase-3 and TUNEL-positive cells was detected in the aSHF (Isl1+) of Hoxb1$^{GoF}$;Mef2c-Cre compared to control embryos, indicating increased apoptosis (*Figure 4N–Q*). Hence, the severe right ventricular hypoplasia resulting from ectopic *Hoxb1* expression was primarily due to extensive cell death and secondarily to reduced proliferation in the aSHF.

To further examine whether over-expression of *Hoxb1* in pSHF cells can lead to heart defects, Hoxb1$^{GoF}$ mice were crossed with Hoxb1$^{IRES-Cre}$ (Hoxb1-Cre) mice (*Bertrand et al., 2011*). In order to better visualize the morphology of the forming heart, whole-mount RNA in situ hybridizations were performed on Hoxb1$^{GoF}$;Hoxb1-Cre and control embryos using a myosin light chain 2 v (Mlc2v) riboprobe (*Figure 4—figure supplement 2A–B*). At E9.5, Hoxb1$^{GoF}$;Hoxb1-Cre hearts were indistinguishable from those of control embryos (*Figure 4—figure supplement 2A–B*). Similarly, gross morphological observations of E13.5 hearts did not reveal any difference in Hoxb1$^{GoF}$;Hoxb1-Cre compared to control embryos (*Figure 4—figure supplement 2C–D*). These data suggest that over-expression of *Hoxb1* in the pSHF does not lead to heart defects, and that the phenotypes observed in the Hoxb1$^{GoF}$;Mef2c-Cre embryos result from misexpression in aSHF progenitor cells that normally do not express *Hoxb1*.

## Mis-expression of *Hoxb1* in the *Mef2c-AHF+* domain disturbs cardiac progenitor identity and blocks cardiac differentiation

To better understand the change of the transcriptional program in the aSHF upon ectopic expression of *Hoxb1*, we performed RNA-seq analysis on dissected progenitor regions from control and Hoxb1$^{GoF}$;Mef2c-Cre embryos (n = 3 each) at E9.5 (16–20 s) (*Figure 5—figure supplement 1A–B*). Read counts showed that *Hoxb1* mRNA was upregulated less than two fold in Hoxb1$^{GoF}$;Mef2c-Cre

compared to control embryos (*Supplementary file 1*). We identified 1378 genes upregulated and 1345 genes downregulated in the SHF of *Hoxb1*[GoF];*Mef2c-Cre* embryos. GO enrichment analysis for the biological processes associated with the upregulated genes showed significant enrichment of the GO terms 'cell death' and 'apoptotic signaling pathway' (*Figure 5B* and *Figure 5—figure supplement 1C–D*), consistent with the decrease of aSHF cell numbers and increase of Cas3+ or TUNEL + cells observed in *Hoxb1*[GoF];*Mef2c-Cre* embryos (*Figure 4N–Q*). These genes included regulators of programmed cell death (*e.g., Bad, Bmf, Trp53,* and *Dapk3*) as well as modulators of growth and RA signaling (*e.g., Gsk3a, Crabp2,* and *Rarα*) (*Figure 5—figure supplement 1D*). GO analysis of the downregulated genes revealed an enrichment in the GO terms 'heart development' and 'muscle cell differentiation' suggesting an inhibition of cardiac differentiation (*Figure 5B* and *Figure 5—figure supplement 1D*). *Fgf10*, a well-characterized marker of the aSHF (*Kelly et al., 2001*), was among the most significant downregulated genes in *Hoxb1*[GoF];*Mef2c-Cre* embryos (p=0.002) (*Figure 5A* and *Figure 5—figure supplement 1D*; *Supplementary file 1*). This finding is consistent with the decrease in *Mlc1v-nlac-24* transgene expression (*Figure 4—figure supplement 1C–G*) and reduction of arterial and venous pole distance measured in *Hoxb1*[GoF];*Mef2c-Cre* embryos (*Figure 4—figure supplement 1H–J*). Among the upregulated genes, we found *Osr1* and *Tbx5*, known to regulate cell cycle progression in the pSHF (*Figure 5C* and *Figure 5—figure supplement 1E*; *Supplementary file 1*; *Zhou et al., 2015*). The upregulation of these genes was confirmed by qPCR and in situ hybridization (*Figure 5D–K* and *Figure 5—figure supplement 1F*). Analysis of *Osr1* and *Tbx5* expression profiles showed an anterior shift of their expression in *Hoxb1*[GoF];*Mef2c-Cre* compared to control embryos (*Figure 5D–K*). Moreover, we found that the pSHF markers *Tbx5* and *Osr1* are both expressed in the *GFP+* cells in *Hoxb1*[GoF];*Mef2c-Cre* embryos, indicating that the ectopic expression of *Hoxb1* in the *Mef2c-AHF+* domain alters cardiac progenitor cell identity (*Figure 5E,E'–K,K''*). Similarly to *Tbx5* and *Osr1*, we observed an anterior shift in expression of the pSHF gene *Nr2f2* in *Hoxb1*[GoF];*Mef2c-Cre* embryos (*Figure 5—figure supplement 1G*).

We further examined whether ectopic expression of *Hoxb1* in the aSHF can alter chromatin accessibility. After micro-dissection and dissociation of the aSHF region from *Hoxb1*[GoF];*Mef2c-Cre* (Hoxb1[GoF]) and *Mef2c-Cre;Rosa*[tdT] (Tomato+) embryos at E9.5 (n = 3 each), Hoxb1[GoF] and Tomato+ regions were used for ATAC-seq (*Figure 5—figure supplement 2A*). We performed a stringent analysis to identify qualitative (present or absent peaks only) differences in chromatin accessibility (*Figure 5—figure supplement 2B*). By comparing the signal for each peak in Tomato+ and Hoxb1-[GoF] samples (in triplicate), we identified 14,348 peaks that were exclusively accessible in the Hoxb1-[GoF] population (*Figure 5—figure supplement 2C*). In order to assess differential chromatin accessibility at each consensus peaks we used an affinity analysis as a quantitative approach (*Figure 5—figure supplement 2D*). ATAC-seq data of Hoxb1[GoF] population revealed a high read count around the promoter regions of *Hoxb1* and *Mef2c* (*Figure 5—figure supplement 2E*). Furthermore, we used HOMER to identify enriched putative transcription factor motifs in the open chromatin regions of the Hoxb1[GoF] population (*Figure 5—figure supplement 2F*). Consistent with the analysis of enriched putative transcription factor motifs in the open chromatin regions of the *Hoxb1*[GFP] (pSHF/GFP+) population (*Figure 3I*), the most enriched motif in open chromatin regions of the Hoxb1[GoF] population was the consensus HOX motif (*Figure 5—figure supplement 2F*). Again, other significantly enriched motifs included PBX motifs. Together these findings suggest that ectopic expression of *Hoxb1* in the *Mef2c-Cre* lineages alters chromatin accessibility of genes involved in the patterning of the SHF.

## Hoxb1 loss of function leads to premature differentiation in the SHF

To complement our functional analysis, we determined if the cardiac differentiation program was affected in the absence of *Hoxb1* function. As previously reported *Hoxb1* mutant embryos have outflow tract defects leading to later anomalies including VSD and mis-alignment of the great arteries (*Roux et al., 2015*). RNA-seq transcriptional profiling was performed on progenitor regions isolated from E8.5 (6–8 s) wild-type and *Hoxb1*-deficient embryos (n = 2 each; *Figure 5L*; *Figure 5—figure supplement 3A–B*). Interestingly, GO term analysis of the upregulated genes revealed a significant enrichment of terms including 'heart development', 'cardiac muscle tissue development' and 'regulation of cell differentiation' (*Figure 5M* and *Figure 5—figure supplement 3C–D*). Upregulation of several myocardial-specific genes (*e.g., Myl2, Myh7, Actn2, Myl3, Nppa (Anf),* and *Nppb (Bnf)*), indicates a premature cardiac differentiation in the SHF of *Hoxb1*[-/-] mutant embryos (*Figure 5N*;

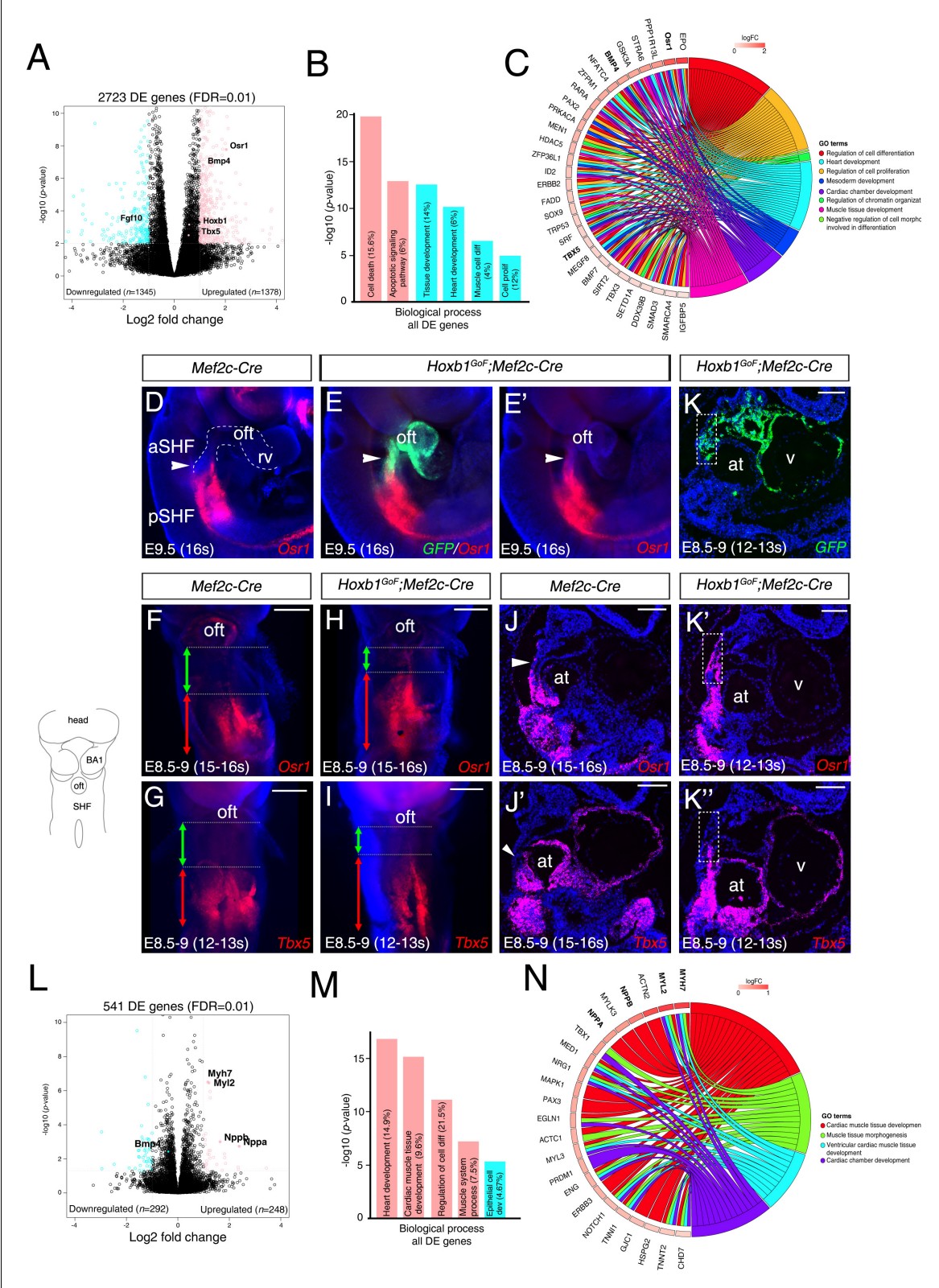

**Figure 5.** Hoxb1 regulates progenitor identity and differentiation in the pSHF. (**A**) Volcano plot of transcriptional profiling results with significantly dysregulated genes between *Hoxb1^GoF^;Mef2c-Cre* and control SHF. The y-axis corresponds to the mean expression value of log₁₀ (*p*-value), and the x-axis displays the log2 fold change value. The colored dots represent the significantly differential expressed transcripts (*p*<0.05); the black dots represent the transcripts whose expression levels did not reach statistical significance (*p*>0.05). Differential expression analysis performed using DESeq2

*Figure 5 continued on next page*

*Figure 5 continued*

revealed 2,723 genes with Log$_2$-fold changes ≥1 at a False Discovery Rate (FDR) ≤0.01. (**B**) Gene ontology (GO) analysis of genes deregulated in *Hoxb1$^{GoF}$;Mef2c-Cre* embryos performed with ClusterProfiler system. The percentage corresponds to the 'BG ratio' obtained in the GO analysis. (**C**) Chord plot showing a selection of genes upregulated in *Hoxb1$^{GoF}$;Mef2c-Cre* embryos present in the represented enriched GO terms. Outer ring shows log2 fold change or GO term grouping (right, key below). Chords connect gene names with GO term groups. (**D**) Whole-mount RNA-FISH for *Osr1* on E9.5 *Mef2c-Cre* embryos in lateral views. (**E, E'**) Whole-mount RNA-FISH for *GFP* (green) *and Osr1* (red) on E9.5 *Hoxb1$^{GoF}$;Mef2c-Cre* embryos. Whole-mount RNA-FISH for *Osr1* in ventral views (**F, H**) and *Tbx5* (**G, I**) on E8.5-9 *Mef2c-Cre* and *Hoxb1$^{GoF}$;Mef2c-Cre* embryos. An anteriorly shifted expression of *Osr1* and *Tbx5* is detected in *Hoxb1$^{GoF}$;Mef2c-Cre* embryos compared to their control littermates (same somite stage – 15-16s for *Osr1* and 12-13s for *Tbx5*). RNA-FISH against *Osr1*, *Tbx5* and *GFP* on serial sections in *Hoxb1$^{GoF}$;Mef2c-Cre* embryos (**K, K', K''**) compared to control littermates (**J, J'**; Serial sagittal sections). (**L**) Volcano plot showing differential expressed genes between *Hoxb1$^{-/-}$* and wild-type samples. The y-axis corresponds to the mean expression value of log$_{10}$ (*p*-value), and the x-axis displays the log2 fold change value. Colored dots represent the significantly differential expressed transcripts (*p*<0.05); the black dots represent the transcripts whose expression levels did not reach statistical significance (*p*>0.05). We identified 249 genes upregulated, and 292 genes downregulated in *Hoxb1$^{-/-}$* embryos. (**M**) GO analysis of genes deregulated in *Hoxb1$^{-/-}$* embryos with ranked by -log$_{10}$ (*p*-value). (**N**) Chord plot showing a selection of genes upregulated in dissected pharyngeal mesoderm of *Hoxb1$^{-/-}$* embryos present in the represented enriched GO terms. Outer ring shows log2 fold change or GO term grouping (right, key below). Chords connect gene names with GO term groups. Nuclei are stained with Hoechst. at, atria; BA1, branchial arch 1; oft, outflow tract; SHF, second heart field; V, ventricle. Scale bars: 200 µm (**F, G, H, I**); 100 µm (**J, J', K, K', K''**).

The online version of this article includes the following figure supplement(s) for figure 5:

**Figure supplement 1.** Quality assessment of RNA-seq data performed on the *Hoxb1$^{GoF}$;Mef2c-Cre* embryos.
**Figure supplement 2.** Chromatin accessibility when Hoxb1 is ectopically expressed in the Mef2c-Cre+ cardiac progenitor cells.
**Figure supplement 3.** Quality assessment of RNA-seq data performed on the *Hoxb1$^{-/-}$* embryos.

*Figure 5—figure supplement 3E*). The upregulation of these myocardial genes was confirmed by qPCR (*Figure 5—figure supplement 3F*). The GO term 'epithelial cell development' was significantly enriched in the downregulated genes (e.g., *Cdh1, Llgl2, and Lrp5*) (*Figure 5M* and *Figure 5— figure supplement 3C–D*), consistent with both the deregulation of the epithelial properties of SHF cells (*Cortes et al., 2018*) and premature differentiation of cardiac progenitor cells (*Soh et al., 2014*). This loss-of-function analysis complements and supports the conclusions of our gain-of-function analysis and identifies a role for Hoxb1 in delaying differentiation and regulating progenitor cell identity in the SHF.

## Abnormal development of the SHF results in AVSD in *Hoxa1$^{-/-}$;Hoxb1$^{-/-}$* embryos

The formation of a transcriptional boundary between arterial and venous pole progenitor cells in the SHF has recently been shown to reflect the dynamic expression of two genes encoding T-box transcription factors, *Tbx1* and *Tbx5* (*De Bono et al., 2018*). Immunofluorescence analysis of E9.5 (22– 23 s) embryos confirmed the complementary expression of Tbx1 and Tbx5 proteins in the SHF (*Figure 6A*). Tbx5 expression is restricted to cells in the pSHF close to the inflow tract, whereas Tbx1 is detected in SHF cells close to the outflow tract. In *Hoxb1$^{GoF}$;Mef2c-Cre* embryos the relative length of the Tbx1+ region close to the outflow tract revealed a significant reduction in the size of the Tbx1+ versus Tbx5+ domains (*Figure 6A,B*), although the boundary between Tbx1 and Tbx5-domains was established. Measurement of the length of each domain revealed that the Tbx1+ domain was significantly reduced in *Hoxb1$^{GoF}$;Mef2c-Cre* compared to *Mef2c-Cre* (control) embryos (*Figure 6C*). In addition, despite a general reduction of the SHF, the Tbx5+ domain is increased in *Hoxb1$^{GoF}$;Mef2c-Cre* compared to control embryos (*Figure 6D*). Therefore, our data are consistent with a shift of the boundary between Tbx1 and Tbx5-domains. We further analyzed the expression of Tbx1 and Tbx5 in *Hoxb1*-mutant embryos. The Tbx5+ domain appears slightly shorter in *Hoxb1$^{-/-}$* embryos than in *Hoxb1$^{+/-}$* littermates (*Figure 6E,F*). Due to redundancy between *Hoxa1* and *Hoxb1* we performed RNA-FISH analysis in compound *Hoxa1$^{-/-}$;Hoxb1$^{-/-}$* embryos. We found that the *Tbx5+* domain was shorter in double *Hoxa1$^{-/-}$;Hoxb1$^{-/-}$* compared to *Hoxb1$^{+/-}$* littermate embryos (*Figure 6G,H*). These experiments confirm that forced activation of *Hoxb1* in the *Mef2c-AHF+* domain perturbs development of the aSHF. We subsequently investigated posterior SHF contributions to the venous pole of *Hoxa1$^{-/-}$;Hoxb1$^{-/-}$* hearts. Characterization of cardiac morphology in

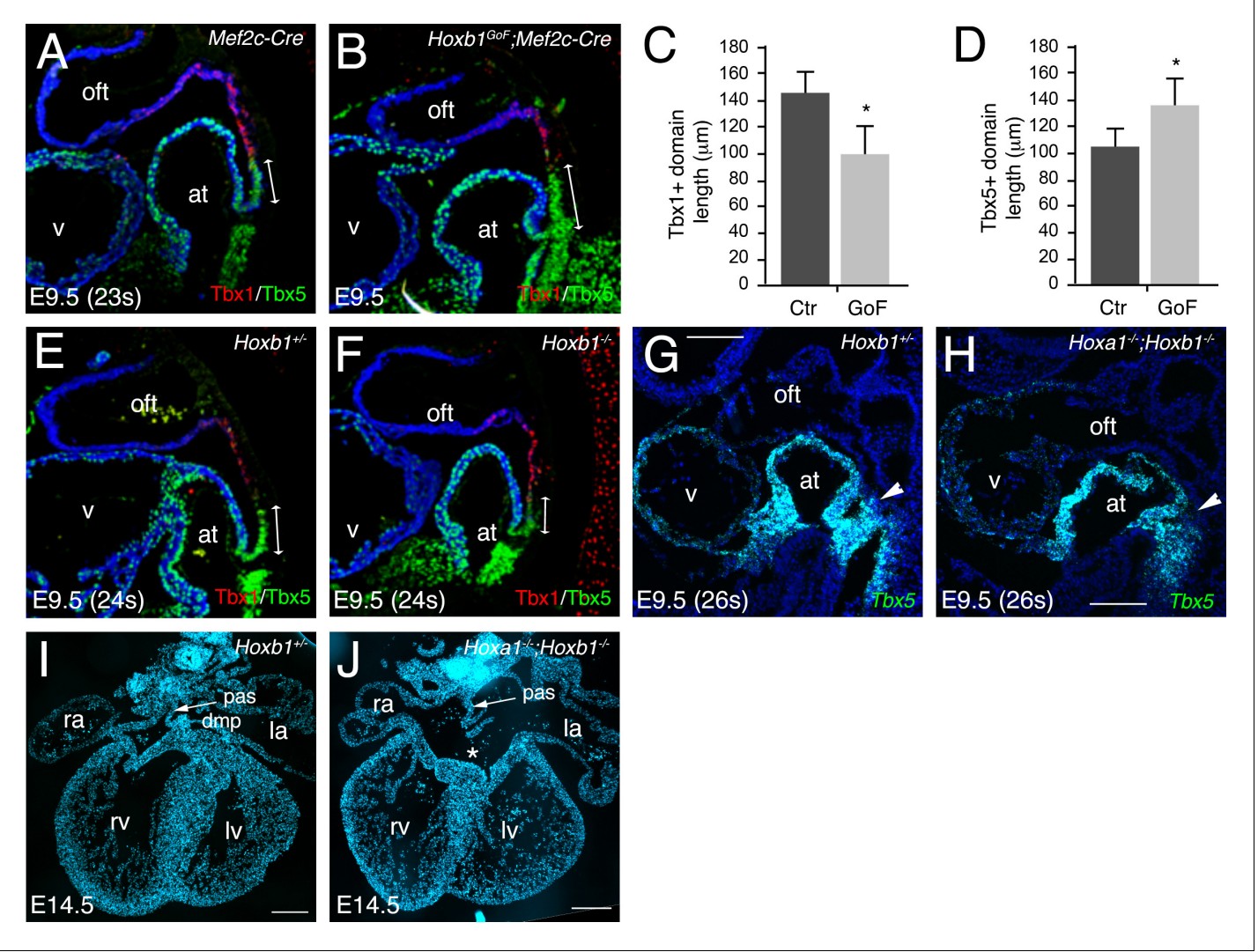

**Figure 6.** *Hoxa1* and *Hoxb1* genes are required for atrioventricular septation. (A–D) Immunofluorescence on medial sagittal sections showing Tbx1 (red) and Tbx5 (green) protein distribution at E9.5 (23–24 s). (A,B) At E9.5, a boundary is observed in *Mef2c-Cre* embryos between Tbx1+ cells close to the arterial pole of the heart and Tbx5+ cells in the pSHF. In *Hoxb1*$^{GoF}$*;Mef2c-Cre* embryos the Tbx1+ domain appears reduced although the boundary is maintained. (C) Measurement of the Tbx1+ domain confirmed a decrease of aSHF length in *Hoxb1*$^{GoF}$*;Mef2c-Cre* (GoF) compared to control (Ctr) embryos. (D) Measurement of the Tbx5+ domain revealed a shift of the boundary since the length of the pSHF was increased in *Hoxb1*$^{GoF}$*;Mef2c-Cre* (GoF) compared to control (Ctr) embryos. (E,F) The Tbx5+ domain appears reduced in *Hoxb1*$^{-/-}$ embryos compared to *Hoxb1*$^{+/-}$ littermates. (G,H) RNA-FISH on sagittal sections showing the reduction of *Tbx5*+ domain in the pSHF of *Hoxa1*$^{-/-}$*;Hoxb1*$^{-/-}$ embryos compared to *Hoxb1*$^{+/-}$ littermates. (I,J) DAPI stained sections of a *Hoxb1*$^{+/-}$ (I) and a *Hoxa1*$^{-/-}$*;Hoxb1*$^{-/-}$ (J) heart at E14.5 showing the primary atrial septum (pas, arrow). Note the AVSD and absence of the DMP in J, n = 3. at, atria; la, left atrium; lv, left ventricle; ra, right atrium; rv, right ventricle; SHF, second heart field; v, ventricle; la, left atrium; lv, left ventricle; ra, right atrium; rv, right ventricle. Scale bars: 100 µm (E,F); 200 µm (G,H).

*Hoxa1*$^{-/-}$*;Hoxb1*$^{-/-}$ hearts at fetal stages revealed lack of the DMP, a posterior SHF derivative, resulting in a primum type atrial septal defect, a class of AVSD (3/3; *Figure 6I,J*). Inappropriate differentiation of SHF cells may contribute to the loss of DMP formation in these mutants, providing the first evidence that *Hoxa1* and *Hoxb1* are required for atrioventricular septation.

## Hoxb1 is a key regulator of cardiac differentiation

We next sought to investigate the function of Hoxb1 upon cardiac induction of mouse embryonic stem (mES) cells (*Figure 7A*). Using a time-course gene expression analysis during cardiac differentiation of mES cells, we detected a peak of *Hoxb1* expression at day 4–5 just before the onset of cardiac differentiation (*Figure 7B*), as determined by the activation of specific cardiac markers such as

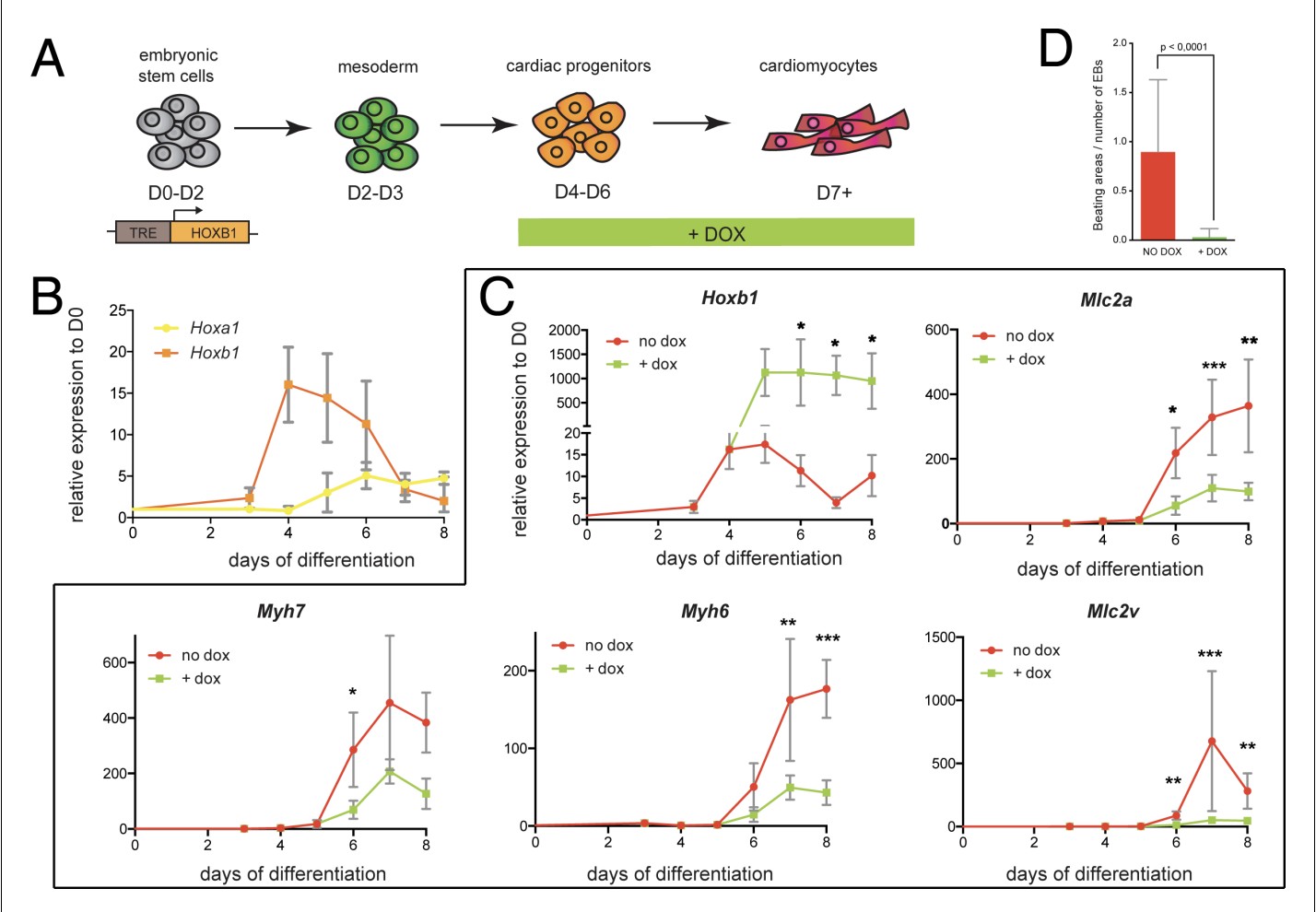

**Figure 7.** *Hoxb1* overexpression in mES leads to arrested cardiac differentiation. (**A**) Scheme of the experiment. Using the *tet-ON/Hoxb1* mouse embryonic stem cell line (***Gouti and Gavalas, 2008***), *Hoxb1* expression was induced by addition of 1 μg/ml of doxycycline (DOX) at the cardiac progenitor stage from day 4 (**D4**) during mES cell differentiation into cardiac cells. (**B**) Kinetics of *Hoxa1* and *Hoxb1* during mES cell differentiation as measured by real time qPCR. Results are normalized for gene expression in undifferentiated mES cells (**D0**). (**C**) Kinetics of expression for *Hoxb1* and *Myh6, Myh7, Mlc2v (or Myl2), Mlc2a* during mES cell differentiation after induction of *Hoxb1* expression (+ dox) or in control condition (no dox). Results are normalized for gene expression in undifferentiated mES cells (**D0**). Paired, one-sided t-test was performed based on relative transcript expression between control (no dox) and doxycycline treatment (+ dox). * indicates a significance level of p<0.05, ** indicates p<0.005, *** indicates p<0.0005. (**D**) Quantification of beating areas relative to the number of embryonic bodies (EBs) at D8 of mES cell differentiation with or without doxycycline addition. Error bars indicate mean + / - SEM; n = 4 experiments.

The online version of this article includes the following figure supplement(s) for figure 7:

**Figure supplement 1.** Expression analysis using the mES cells overexpressing model.

**Figure supplement 2.** Arrested cardiac differentiation in mES cells using a lower induction of Hoxb1.

*Myh6, Myh7, Mlc2a* and *Mlc2v* (***Figure 7—figure supplement 1A***). Consistent with the initial activation of *Hoxb1* and *Hoxa1* during heart development in the mouse (***Bertrand et al., 2011***), the peak of *Hoxa1* expression was detected after day 5 (***Figure 7B***).

Next, we challenged the system by inducing continuous *Hoxb1* overexpression using the mES[Tet-on/Hoxb1] line (***Gouti and Gavalas, 2008***). Permanent DOX (1 or 0.2 μg/ml) treatment from day 4 of direct cardiac induction onwards (***Figure 7C*** and ***Figure 7—figure supplement 2***) interfered with the differentiation process of cardiac cells as shown by specific downregulation of the expression of *Myh7, Myh6, Mlc2a,* and *Mlc2v* (***Figure 7C*** and ***Figure 7—figure supplement 2***). Accordingly, in

these conditions, we found a decreased number of beating embryoid bodies (EBs) (*Figure 7D* and *Figure 7—figure supplement 2*). Absence of upregulation of cell death markers attested that reduction of beating EBs was not caused by ES cell apoptosis (*Figure 7D* and *Figure 7—figure supplement 1C*). Interestingly, we also observed an upregulation of *Osr1*, *Tbx5* and *Bmp4* expression under these conditions, suggesting that the cellular identity of differentiating EB cells is changed, consistent with our in vivo observations (*Figure 5C–K* and *Figure 7—figure supplement 1D*). Therefore, our data suggest that *Hoxb1* activation in mES cells results in arrest of cardiac differentiation and failure of proper identity commitment, consistent with in vivo results.

## Hoxb1 represses the expression of the differentiation marker *Nppa*

To identify how Hoxb1 controls cardiac differentiation, we analyzed the regulation of *Nppa* and *Nppb* expression, two markers of chamber-specific cardiomyocytes (*Houweling et al., 2005*). RNA-seq data showed a higher read count for *Nppa* and *Nppb* in the *Hoxb1$^{-/-}$* compared to control embryos (*Figure 8A*). At E9.5, RNA-FISH confirmed an ectopic expression of *Nppa* in the SHF of *Hoxb1$^{-/-}$* (*Figure 8B,C*) and *Hoxa1$^{-/-}$Hoxb1$^{-/-}$* (*Figure 8D,E*) embryos. In contrast, upon *Hoxb1* induction, the expression of *Nppa* and *Nppb* was downregulated in EBs (*Figure 7—figure supplement 1E*). Therefore, we hypothesized that *Nppa* and *Nppb* may be negatively regulated by Hoxb1 in the pSHF. ChIP-seq data in mouse ES cell lines had shown that Hoxa1, a paralog of Hoxb1, and HDAC-1 and −2 bind the *Nppa* and *Nppb* loci (*De Kumar et al., 2017*; *Whyte et al., 2012*; *Figure 8G*). Coherent with these observations, we found potential HOX binding sites in a 0.7 kb *Nppa* fragment previously shown to be responsible for the expression of the *Nppa* in the developing heart (*Habets et al., 2002*). Thus, we hypothesized that *Nppa* and *Nppb* may be a direct Hoxb1 target genes in the pSHF. As described (*Durocher et al., 1997*), transfection of Nkx2-5 alone or co-transfection of Nkx2-5 and Gata4 resulted in strong activation of the 0.7 kb *Nppa* promoter containing an HOX-motif in both Cos-7 and NIH3T3 cells (*Figure 8I* and *Figure 8—figure supplement 1A*). However, this activity decreased three-fold upon co-transfection with a Hoxb1 expression vector or co-transfection of Hoxb1 and Hoxa1, which are co-expressed in pSHF progenitor cells in vivo (*Figure 8I*), demonstrating the repressive role of Hoxb1 on the 0.7 kb *Nppa* promoter. We next assessed the activity of the 0.7 kb *Nppa* promoter in cells treated with trichostatin-A (TSA) an inhibitor of the HDAC activity known to regulate HOX functions (*McKinsey, 2012*). HDACs inhibition increased the luciferase activity of the reporter constructs (*Figure 8J* and *Figure 8—figure supplement 1B*). When co-transfected, Hoxb1 or Hoxa1 suppressed the TSA-mediated activation of the *Nppa* promoter in cell culture (*Figure 8J*). We further examined whether Hoxb1 can also repress the transcription of another myocardial gene, *Tnnt2* (cardiac troponin T, *cTnT*). As for *Nppa* and *Nppb*, RNA-seq data showed a higher read count for *Tnnt2* in the *Hoxb1$^{-/-}$* compared to control embryos (*Figure 8F*). Analysis of ChIP-seq dataset showed that Hoxb1 as well as Meis1 bind to the *Tnnt2* locus (*Figure 8H*). We identified potential HOX binding sites in the promoter of the rat *cTnT* promoter (0.5 kb), which contains also MEF2 binding sites (*Wang et al., 1994*). We assessed the activity of the 0.5 kb *Tnnt2* promoter in cells treated with TSA. As in the case of the *Nppa* promoter we observed that co-transfection with a Hoxb1 expression vector or co-transfection of Hoxb1 and Hoxa1 suppressed TSA-medicated activation of the *Tnnt2* promoter in Cos7 cells (*Figure 8J*). Together these results suggest that Hoxb1 inhibits differentiation in the pSHF by directly repressing myocardial gene transcription even under conditions of histone acetylation.

## Discussion

In this study, we characterize the transcriptional profile of subpopulations of SHF progenitor cells contributing to the forming heart and identify central roles of Hoxb1 in the posterior SHF. We report that forced activation of *Hoxb1* in the *Mef2c-AHF* lineage results in a hypoplastic right ventricle and show that Hoxb1 has a dual role in activating the posterior program of the SHF and inhibiting premature cardiac progenitor differentiation through the transcriptional repression of myocardial genes. Thus, Hoxb1 coordinates patterning and deployment of SHF cells during heart tube elongation and altered Hoxb1 expression contributes to CHD affecting both poles of the heart tube.

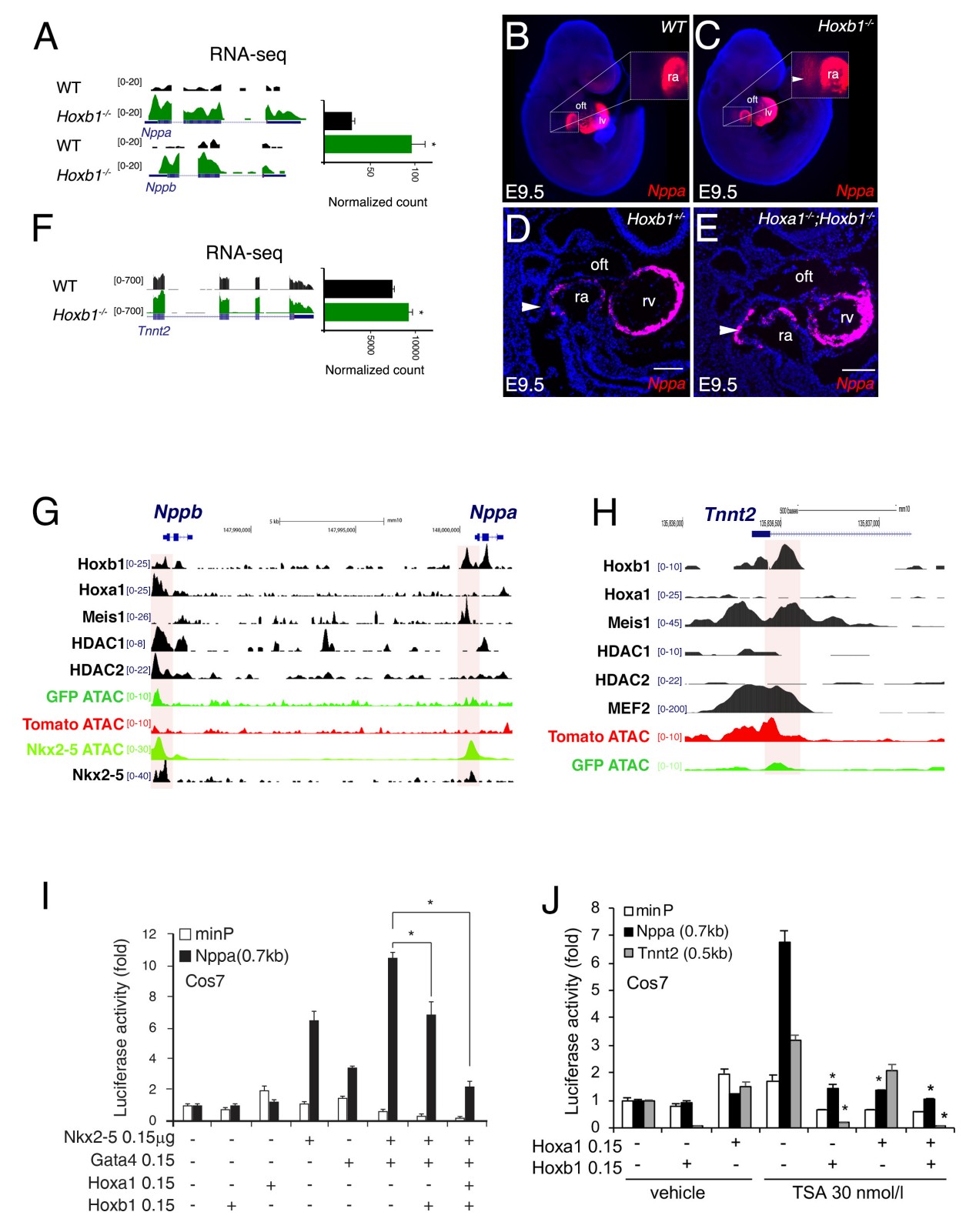

**Figure 8.** Hoxb1 regulates cardiac differentiation through transcriptional repression of myocardial genes. (**A**) Browser views of *Nppa* and *Nppb* gene loci with RNA-seq profiles of *Hoxb1⁻/⁻* (green) and wild-type (WT) population (black). Data represent the merge of technical and biological replicates for each cell type. The *y*-axis scales range from 0 to 20 in normalized arbitrary units. (**B,C**) Whole mount RNA-FISH for *Nppa* in WT (**B**) and *Hoxb1⁻/⁻* (**C**) E9.5 embryos. Inset displays higher magnification of the posterior heart region. Nuclei are stained with Dapi. RNA-FISH on sagittal sections for *Nppa* in
*Figure 8 continued on next page*

Figure 8 continued

*Hoxb1*[+/-] (D) and *Hoxa1*[-/-];*Hoxb1*[-/-] (E) at stage E9.5. (F) Browser view of *Tnnt2* locus with RNA-seq profiles of *Hoxb1*[-/-] (green) and wild-type (WT) population (black). Data represent the merge of technical and biological replicates for each cell type. The *y*-axis scales range from 0 to 700 in normalized arbitrary units. (G,H) Browser view of *Nppa*, *Nppb* (G) and *Tnnt2* (H) loci with ATAC-seq on purified cardiac cells and ChIP-seq profiles of Hoxb1, Hoxa1 (*De Kumar et al., 2017*), Meis1 (*Losa et al., 2017*), Nkx2-5 (*van den Boogaard et al., 2012*), Mef2 (*Akerberg et al., 2019*), HDAC1 and HDAC2 (*Whyte et al., 2012*). (I) Constructs were co-transfected with Nkx2-5, Gata4, Hoxa1 and Hoxb1 expression vectors into Cos-7 cells. Luciferase activity was determined and normalized as fold over the reporter alone (mean ± SEM, n = 3, *p<0.05 for Nkx2-5 and Hoxb1 *versus* Nkx2-5, using ANOVA). (J) Luciferase reporter assays on the −633/+87 bp region of the *Nppa* promoter and the −497/+1 bp region of the *Tnnt2* promoter. Cos-7 cells co-transfected with Hoxa1 and Hoxb1 expressing vector or not were treated in the absence or presence of 30 nmol/l TSA. Bars represent mean ± SEM (n = 3). Statistical test was conducted using ANOVA (*p<0.05 for Hoxa1, Hoxb1 and TSA treatment versus Hoxb1 or TSA treatment). oft, outflow tract; lv, left ventricle; ra, right atrium.

The online version of this article includes the following figure supplement(s) for figure 8:

**Figure supplement 1.** The 0.7 kb *Nppa* promoter is a functional target for Hoxb1.

## Transcriptional profile and chromatin accessibility mapping of the posterior SHF

Here, we present the first analysis of open chromatin in purified SHF progenitor cells using the ATAC-seq method. Such datasets are important to understand the tightly regulated genetic networks governing heart development and to determine how these networks become deregulated in CHD. Integration with transcriptomes of sorted cardiac progenitor cells led us to the discovery of novel pSHF markers illustrating how this study provides a large dataset and multiple new avenues of investigation for the future. Among genes enriched in the *Hoxb1*-expressing progenitors we found several genes known to be important for the inflow tract development, including *Tbx5*, *Osr1*, *Foxf1*, *Bmp4*, *Wnt2*, and *Gata4*. Importantly, *Tbx5*, *Osr1*, *Foxf1*, and *Bmp4*, have also been implicated in DMP, which derives from both *Hoxb1* and *Mef2c-AHF*-lineages, (*Zhou et al., 2017*; *Burns et al., 2016*; *Zhou et al., 2015*; *Hoffmann et al., 2014*; *Briggs et al., 2013*; *Xie et al., 2012*). Our analysis of the double positive *Hoxb1*[GFP] (GFP+) and *Mef2c-Cre;Rosa*[tdT] (Tomato+) population at E9.5 showed an activation of *Osr1* and *Adlh1a2* transcripts, whereas both *Hoxb1* and *Tbx5* transcripts are weakly expressed (*Figure 2K*). These results suggest a progressive change in cell identity during SHF deployment to alternate cardiac poles.

## Hox genes are required for atrioventricular septation

Although recent single-cell RNA-seq analyses identified specific pSHF and aSHF clusters, the subpulmonary and DMP specific sub-populations have not yet been characterized, probably due to the restricted number of these progenitors (*de Soysa et al., 2019*; *Pijuan-Sala et al., 2019*). The DMP protrudes into the atrial lumen to contribute to the atrioventricular mesenchymal complex and muscular base of the atrial septum. Perturbation of DMP development in embryos mutant for transcription factors and signaling molecules regulating posterior SHF deployment results in AVSDs (*De Bono et al., 2018*; *Rana et al., 2014*; *Briggs et al., 2012*; *Xie et al., 2012*; *Goddeeris et al., 2008*). We observed that either *Hoxb1* or *Hoxa1;Hoxb1* loss of function leads to premature differentiation in the SHF. Importantly, premature myocardial differentiation in the SHF is known to contribute to defective DMP development and leads to AVSD (*Goddeeris et al., 2008*). Consistent with these observations we found AVSD with absence of the DMP in *Hoxa1*[-/-];*Hoxb1*[-/-] mutant embryos, identifying Hox genes as upstream players in the etiology of this common form of CHD. Among the pSHF genes upregulated in *Hoxb1*[GoF];*Mef2c-Cre* and downregulated in *Hoxb1*[-/-] mutant embryos were *Bmp4* and *Gata4* (*Figure 5*). Bmp4 is expressed in the DMP (*Briggs et al., 2016*; *Burns et al., 2016*; *Sun et al., 2015*; *Briggs et al., 2013*) and mutations in BMP4 and the BMP receptor Alk2 have been implicated in atrial septal defects and AVSDs (*Smith et al., 2011*). Similarly, Gata4 is required for DMP development and atrioventricular septation (*Liao et al., 2008*, *Zhou et al., 2017*).

## Hoxb1 is required to repress cardiac differentiation in the pSHF

Examination of ectopic expression of *Hoxb1* in the *Mef2c-Cre* derivatives showed a hypoplastic right ventricle phenotype. RNA-seq analysis showed that ectopic expression of *Hoxb1* results in mis-specification of the anterior program leading to the downregulation of genes involved in cardiac differentiation. Consistent results were observed in *Hoxb1*$^{-/-}$ mutant embryos where GO analysis showed upregulated genes related to 'cardiac muscle tissue development' and 'muscle system process'. This aberrant gene program prompted us to hypothesize that Hoxb1 blocks the differentiation process by inhibiting activation of a set of myocardial genes. Our observations in the mES cell system confirmed the repressive function of Hoxb1 during cardiac differentiation. Consistent with our previous work, the present study suggests that *Hox* genes delay or repress the differentiation of cardiac progenitor cells, thus allowing them to participate in heart formation at the right time and coordinating progressive heart tube elongation. Interestingly, a small number of apoptotic cells were found to be present in the aSHF in *Hoxb1*$^{GoF}$;*Mef2c-Cre* embryos (*Figure 4N–Q*). However, although we observed downregulation of cardiac differentiation markers after Dox treatment of mES$^{Tet-on/Hoxb1}$ cells, no activation of the cell death program was detected in the beating EBs. These observations suggest a difference between the in vivo and in vitro models, which may be linked to culture conditions used for mES cells. Therefore, we speculate that the cell death within the aSHF could be due to lack of survival signals. Such survival signals may include FGFs since increase apoptosis of Isl1+ cardiac progenitor cells was reported in the double homozygous *Fgf8;Fgf10* mutants (*Watanabe et al., 2010*), and conditional *Fgf8*$^{flox/-}$ mutants (*Park et al., 2006*). Accordingly, our RNA-seq analysis revealed a significant downregulation of *Fgf10* mRNA level in *Hoxb1*$^{GoF}$;*Mef2c-Cre* embryos (*Figure 5A*; p=0.005). In vitro analysis in conjunction with our RNA-seq analysis suggest that Hoxb1 may act as transcriptional repressor of structural myocardial genes. Hoxb1 activity on *Nppa* and *Tnnt2* promoters was consistent with our in vivo analysis. The 0.7 kb *Nppa* fragment is responsible for the developmental expression pattern of the *Nppa* gene and is itself regulated by Gata4 and Nkx2-5 (*Habets et al., 2002*). Our data provide molecular evidence that Hoxb1 functions to repress differentiation in the pSHF. Several studies have suggested that Hoxb1 plays a repressive role in differentiation of other cell types (*Chen et al., 2012*; *Bami et al., 2011*). Indeed, HOX transcription factors can function as activators or repressors (*Mann et al., 2009*; *Saleh et al., 2000*) raising the question of how gene activation versus repression is determined. Our analysis of ATAC-seq in pSHF population demonstrated overrepresentation of specific binding motifs. Several transcription factors known to bind to these motifs, including Nkx2-5, Gata4, Pbx1/2/3 and Meis1/2, have been previously associated with cardiac differentiation. Nkx2-5 cooperates with Hox factors to regulate the timing of cardiac mesoderm differentiation (*Behrens et al., 2013*). TEAD proteins regulate a variety of processes including cell proliferation survival and heart growth (*Lin et al., 2016*; *von Gise et al., 2012*). Unexpectedly, our results implicate TEAD proteins in activating regulatory genes and potentially promoting the proliferation/outgrowth of cardiac progenitors in the pSHF, where nuclear accumulation of the TEAD partners YAP/TAZ has been reported (*Francou et al., 2017*). The HOX protein family has been previously shown to interact other with homeodomain proteins, including the TALE-class Pbx and Meis family members (*Lescroart and Zaffran, 2018*). This further supports the conclusion that Hoxb1 uses cofactors to activate or repress cardiac genes. Repression is mediated by direct recruitment of repressor complexes, such as NuRD, or through maintenance of repressed chromatin states, such as those mediated by Polycomb complexes functioning in an HDAC complex (*Schuettengruber and Cavalli, 2009*). Several HOX proteins reportedly bind HAT or HDACs enzymes (*Ladam and Sagerström, 2014*; *Saleh et al., 2000*). Meis proteins promote HAT recruitment by displacing HDACs to permit HAT binding (*Shen et al., 2001*). Our results suggest that Meis proteins might thus modulate HAT/HDAC accessibility at Hox-regulated regulatory sequences to delay differentiation in the SHF. Indeed, a recent study using live imaging of cell lineage tracing and differentiation status suggests that there is a discrete temporal lag between the first and second waves of differentiation that form the mouse heart (*Ivanovitch et al., 2017*). Such a delay of differentiation is essential to orchestrate early cardiac morphogenesis. Hoxb1 may thus contribute to controlling this differentiation delay, in particular through maintaining pSHF progenitor cells in an undifferentiated state until they are added to the venous pole or the inferior wall of the outflow tract. Future work will define how *Hoxb1* expression is downregulated to release the cardiac differentiation process during SHF deployment.

# Materials and methods

## Key resources table

| Reagent type (species) or resource | Designation | Source or reference | Identifiers | Additional information |
|---|---|---|---|---|
| Strain, strain background *Mus musculus* | C57Bl/6J | Charles River | | |
| Genetic reagent *Mus musculus* | Hoxb1[GFP] | PMID:11076756 | RRID:MGI:1928090 | Mario Capecchi (HHMI) |
| Genetic reagent *Mus musculus* | Hoxb1[IRES-Cre] | PMID:12815623 | RRID:MGI:2668513 | Mario Capecchi (HHMI) |
| Genetic reagent *Mus musculus* | Tg(CAG-Hoxb1-EGFP)[1Sza] | PMID:30134070 | | |
| Genetic reagent *Mus musculus* | Mef2c-AHF-Cre | PMID:15253934 | RRID:MGI:4940068 | Brian Black (UCSF) |
| Genetic reagent *Mus musculus* | Gt(ROSA)26Sor[tm1Sor] (R26R) | PMID:9916792 | RRID:MGI:1861932 | |
| Genetic reagent *Mus musculus* | Gt(ROSA)26 Sor[tm9(CAG-tdTomato)Hze] | PMID:20023653 | RRID:MGI:3809523 | |
| Genetic reagent *Mus musculus* | Mlc1v-nlacZ-24 or Fgf10[Tg(Myl3-lacZ)24Buck] | PMID:15217909 | RRID:MGI:3629660 | |
| Genetic reagent *Mus musculus* | Mlc3f-nlacZ-2[E] or Tg(Myl1-lacZ)1Ibdml | PMID:15217909 | RRID:MGI:5449224 | |
| Gene (*Mus musculus*) | Hoxb1 | *Mus musculus* genome resource | RRID:MGI:96182 | |
| Antibody | Rabbit anti-GFP Polyclonal Antibody | Thermo Fisher Scientific | Cat# A-11122, RRID:AB_221569 | IF (1:500) |
| Antibody | Rabbit anti-Hoxb1 Polyclonal Antibody | Covance | Cat# PRB-231P-100, RRID:AB_291592 | IF (1:200) |
| Antibody | Mouse anti-αactinin Monoclonal Antibody | Sigma | A7732, RRID:AB_2221571 | IF (1:500) |
| Antibody | Mouse anti-myosin Monoclonal Antibody | DHSB | Cat# MF 20, RRID:AB_2147781 | IF (1:100) |
| Antibody | Rabbit anti- cleaved Caspase3 Polyclonal Antibdoy | Cell signaling Technology | Cat# 9661, RRID:AB_2341188 | IF (1:300) |
| Antibody | Rabbit anti-phospho-Histone H3 Polyclonal Antibody | Millipore | Cat# 06–570, RRID:AB_310177 | IF (1:400) |
| Antibody | Mouse anti-Islet1 Monoclonal Antibody | DSHB | Cat# 39.4D5, RRID:AB_2314683 | IF (1:100) |
| Antibody | Rabbit anti-Tbx1 Polyclonal Antibody | LifeSpan | Cat# LS-C31179-100, RRID:AB_911118 | IF (1:100) |
| Antibody | Goat anti-Tbx5 Polyclonal Antibody | Santa Cruz | Cat# sc-17866, RRID:AB_2200827 | IF (1:250) |
| Antibody | Donkey anti-rat IgG (H+L) Alexa Fluor 488 | Thermo Fisher Scientific | Cat# A-21208, RRID:AB_141709 | IF (1:500) |
| Antibody | Donkey anti-Mouse IgG (H+L) Alexa Fluor 555 | Thermo Fisher Scientific | Cat# A-31570, RRID:AB_2536180 | IF (1:500) |
| Antibody | Goat anti-rabbit IgG (H+L) Alexa Fluor 647 | Thermo Fisher Scientific | Cat# A-21244, RRID:AB_2535812 | IF (1:500) |
| Sequence-based reagent | mm-Hoxb1-C1 *Mus musculus* | Acdbio | 541861 | RNA-FISH probe |
| Sequence-based reagent | mm-Nppa *Mus musculus* | Acdbio | 418691 | RNA-FISH probe |

*Continued on next page*

*Continued*

| Reagent type (species) or resource | Designation | Source or reference | Identifiers | Additional information |
|---|---|---|---|---|
| Sequence-based reagent | mm-Bmp4-C1 *Mus musculus* | Acdbio | 401301 | RNA-FISH probe |
| Sequence-based reagent | mm-Gata4 *Mus musculus* | Acdbio | 417881 | RNA-FISH probe |
| Sequence-based reagent | mm-Aldh1a2 *Mus musculus* | Acdbio | 447391 | RNA-FISH probe |
| Sequence-based reagent | mm-eGFP-C3 | Acdbio | 400281 | RNA-FISH probe |
| Sequence-based reagent | mm-Isl1 | Acdbio | 451931 | RNA-FISH probe |
| Sequence-based reagent | mm-Nr2f2-C3 *Mus musculus* | Acdbio | 480301 | RNA-FISH probe |
| Sequence-based reagent | mm-tdTomato-C3 | Acdbio | 317041 | RNA-FISH probe |
| Sequence-based reagent | mm-Tbx1-C1 *Mus musculus* | Acdbio | 481911 | RNA-FISH probe |
| Sequence-based reagent | mm-Tbx5-C2 *Mus musculus* | Acdbio | 519581 | RNA-FISH probe |
| Sequence-based reagent | mm-Osr1-C2 *Mus musculus* | Acdbio | 496281 | RNA-FISH probe |
| Commercial assay or kit | RNAscope Multiplex Fluorescent v2 Assay | Acdbio | 323110 | |
| Commercial assay or kit | Ovation RNAseq v2 kit | NuGEN | 7102–08 | RNA-seq |
| Commercial assay or kit | AffinityScript Multiple Temperature cDNA synthesis kit | Agilent | 600107 | cDNA synthesis kit |
| Commercial assay or kit | Master Mix PCR Power SYBR Green | ThermoFisher | 4334973 | qPCR |
| Commercial assay or kit | NucleoSpin RNA XS | Machery-Nagel | 740902.50 | RNA extraction |
| Recombinant DNA reagent | pNppa(−633/+87)-Luc (plasmid) | PMID:12023302 | | Vincent Christoffels (Amsterdam University Medical Center) |
| Recombinant DNA reagent | pTnnT2(−497/+1)-Luc (plasmid) | PMID:12023302 | | Vincent Christoffels (Amsterdam University Medical Center) |
| Cell line (Cercopithecus aethiops) | Cos-7 | ATCC | RRID:CVCL_0224 CRL-1651 | |
| Cell line *Mus musculus* | NIH3T3 | ATCC | RRID:CVCL_0594 CRL-1658 | |
| Cell line *Mus musculus* | mES^Tet-On/Hoxb1 | PMID:18499896 | | Anthony Gavalas (Technische Universitat Dresden) |
| Sequence-based reagent | Bmp4-FW | This paper | qPCR primers | TTCCTGGTAACCGAATGCTGA |
| Sequence-based reagent | Bmp4-Rev | This paper | qPCR primers | CCTGAATCTCGGCGACTTTTT |
| Sequence-based reagent | Bag3-Fw | This paper | qPCR primers | GCCCTAAGGACACTGCATCTT |
| Sequence-based reagent | Bag3-Rev | This paper | qPCR primers | GCTGGGAGTAGGCATGGAAA |
| Sequence-based reagent | Bak1-Fw | This paper | qPCR primers | CCTTCTGAACAGCAGGTTGC |

*Continued on next page*

*Continued*

| Reagent type (species) or resource | Designation | Source or reference | Identifiers | Additional information |
|---|---|---|---|---|
| Sequence-based reagent | Bak1-Rev | This paper | qPCR primers | GACCCACCTGACCCAAGATG |
| Sequence-based reagent | Dapk1-Fw | This paper | qPCR primers | GAGGTGGTGGCTGCGTC |
| Sequence-based reagent | Dapk1-Rev | This paper | qPCR primers | CGCAGACCTCCGGTCC |
| Sequence-based reagent | GFP-Fw | This paper | qPCR primers | CGACGTAAACGGCCACAAGTT |
| Sequence-based reagent | GFP-Rev | This paper | qPCR primers | TTGATGCCGTTCTTCTGCTTGT |
| Sequence-based reagent | Hoxa1-Fw | This paper | qPCR primers | AGAAACCCTCCCAAAACAGG |
| Sequence-based reagent | Hoxa1-Rev | This paper | qPCR primers | TTGTTGAAGTGGAACTCCTTCTC |
| Sequence-based reagent | Hoxb1-Fw | This paper | qPCR primers | AAGAGAAACCCACCTAAGACAGC |
| Sequence-based reagent | Hoxb1-Rev | This paper | qPCR primers | TGAAGTTTGTGCGGAGACC |
| Sequence-based reagent | Hprt-Fw | This paper | qPCR primers | TGTTGGATATGCCCTTGACT |
| Sequence-based reagent | Hprt-Rev | This paper | qPCR primers | GATTCAACTTGCGCTCATCT |
| Sequence-based reagent | Isl1-Fw | This paper | qPCR primers | GCAACCCAACGACAAAACTAA |
| Sequence-based reagent | Isl1-Rev | This paper | qPCR primers | CCATCATGTCTCTCCGGACT |
| Sequence-based reagent | Nppa-Fw | This paper | qPCR primers | CACAGATCTGATGGATTTCAAGA |
| Sequence-based reagent | Nppa-Rev | This paper | qPCR primers | CCTCATCTTCTACCGGCATC |
| Sequence-based reagent | Nppb-Fw | This paper | qPCR primers | GTCCAGCAGAGACCTCAAAA |
| Sequence-based reagent | Nppb-Rev | This paper | qPCR primers | AGGCAGAGTCAGAAACTGGA |
| Sequence-based reagent | Nr2f2-Fw | This paper | qPCR primers | CCTCAAAGTGGGCATGAGAC |
| Sequence-based reagent | Nr2f2-Rev | This paper | qPCR primers | TGGGTAGGCTGGGTAGGAG |
| Sequence-based reagent | Osr1F-Fw | This paper | qPCR primers | AGAAGCGTCAGAAGTCTAGTTCG |
| Sequence-based reagent | Osr1R-Rev | This paper | qPCR primers | GGAACCGCAATGATTTCAA |
| Sequence-based reagent | Parp1-Fw | This paper | qPCR primers | AGGGCTGCCTGGAGAAGATA |
| Sequence-based reagent | Parp1-Rev | This paper | qPCR primers | TCGTCCCGCTTCTTGACAAA |
| Sequence-based reagent | Raldh2-Fw | This paper | qPCR primers | CATGGTATCCTCCGCAATG |
| Sequence-based reagent | Raldh2-Rev | This paper | qPCR primers | GCGCATTTAAGGCATTGTAAC |
| Sequence-based reagent | Tbx5-Fw | This paper | qPCR primers | CCCGGAGACAGCTTTTATCG |

*Continued*

| Reagent type (species) or resource | Designation | Source or reference | Identifiers | Additional information |
|---|---|---|---|---|
| Sequence-based reagent | Tbx5-Rev | This paper | qPCR primers | TGGTTGGAGGTGACTTTGTG |
| Sequence-based reagent | Trp53-Fw | This paper | qPCR primers | AGTATTTCACCCTCAAGATCCGC |
| Sequence-based reagent | Trp53-Rev | This paper | qPCR primers | GGAGCTAGCAGTTTGGGCTT |

## Mice

All animal procedures were carried out under protocols approved by a national appointed ethical committee for animal experimentation (Ministère de l'Education Nationale, de l'Enseignement Supérieur et de la Recherche; Authorization N°32–08102012). The $Hoxb1^{GFP}$, $Hoxb1^{IRES-Cre}$ alleles and the $Tg(CAG-Hoxb1-EGFP)^{1Sza}$ transgene ($Hoxb1^-$ and $Hoxb1^{GoF}$ respectively) have been previously described (Zaffran et al., 2018; Gaufo et al., 2000). The reporter lines $Gt(ROSA)26Sor^{tm1Sor}$ (R26R), $Gt(ROSA)26Sor^{tm9(CAG-tdTomato)Hze}$ ($Rosa^{tdTomato}$) have been previously described (Madisen et al., 2010; Soriano, 1999). The Mef2c-AHF-Cre (AHF: Anterior Heart Field; Mef2c-Cre), Mlc1v-nlacZ-24 and Mlc3f-nlacZ-2E mice have been previously described (Verzi et al., 2005; Zaffran et al., 2004; Kelly et al., 2001). $Hoxb1^{GoF/+}$ mice were maintained on a C57Bl/6 background and inter-crossed with Mef2c-Cre with or without Mlc1v24 to generate compound heterozygous embryos at a Mendelian ratio.

## Cell culture

Mouse $ES^{Tet-On/Hoxb1}$ ES lines were generated by the Gavalas laboratory (Gouti and Gavalas, 2008). ES cells were cultured on primary mouse embryonic fibroblast feeder cells. ES cells medium was prepared by supplementing GMEM-BHK-21 (Gibco) with 7.5% FBS, 1% non-essential amino acids, 0.1 mM beta-mercaptoethanoland LIF conditioned medium obtained from pre-confluent 740 LIF-D cells that are stably transfected with a plasmid encoding LIF (Zeineddine et al., 2006). For cardiac differentiation, ES cells were re-suspended at $2.5 \times 10^4$ cells/ml in GMEM medium supplemented with 20% fetal calf serum, 1% non-essential amino-acids, and 0.1 mM beta-mercaptoethanol in hanging drops (22 µl) plated on the inside lids of bacteriological dishes. After 48 hr EBs were transferred in 10 ml medium to 10 cm bacteriological dishes. At day 5 EBs were plated on tissue culture dishes coated with gelatin, allowed to adhere. Expression of Hoxb1 was induced by addition of doxycycline (DOX) (Sigma - 1 or 0.2 µg/ml) from day four to the end of the experiment. The medium was changed every two days.

## Cell sorting

E9.5 (16s) transgenic progenitor heart regions were dissected, pooled (n > 3 embryos for each genotype) and digested with 0.25% Trypsin/EDTA (Invitrogen), neutralized in DMEM (Invitrogen) containing 5% FBS and 10 mmol/L HEPES (Invitrogen), rinsed and resuspended in PBS, and passed through a 70 mm nylon cell strainer (Falcon). Samples were sorted on a FacsAria flow cytometer (BD) using FACSDiva 8.0.1 software. Samples were gated to exclude debris and cell clumps. The number of E9.5 $Hoxb1^{GFP}$ progenitor cells and $Mef2c-Cre;Rosa^{tdT}$ progenitor cells per embryo obtained were typically 600 to 900, respectively. Fluorescent cells were collected into PBS and processed for RNA extraction or ATAC-seq.

## Histological and immunostaining

Standard histological procedures were used (Roux et al., 2015). Heart from $Hoxb1^{GoF}$;Mef2c-Cre and littermate controls were fixed in neutral-buffered 4% paraformaldehyde in PBS, rinsed, dehydrated, paraffin-embedded and tissue sections cut at 8 µm. Sections were stained with Harris' hematoxylin and eosin (H and E) (Sigma). For immunostaining embryos from $Hoxb1^{-/-}$ or $Hoxb1^{GoF}$;Mef2c-Cre and littermate controls were fixed at 4°C for 20 min in 4% paraformaldehyde, rinsed in PBS, equilibrated to 15% sucrose and embedded in O.C.T. Cryo-sections were cut at 12 µm, washed in PBS and pre-incubated in blocking solution (1%BSA, 1% Serum, 0.2% Tween20 in PBS). Primary

antibodies were applied overnight at 4℃, followed by secondary detection using Alexa Fluor conjugated (Molecular Probes) secondary antibodies. Sections were photographed using an AxioImager Z2 microscope (Zeiss) and photographed with an Axiocam digital camera (Zen 2011, Zeiss).

The following primary antibodies were used in this study: rabbit anti-Hoxb1 (Covance; 1/200), rabbit anti-GFP (Life Technologies; 1/500), mouse anti-αactinin (sigma; 1/500), mouse anti-MF-20 (DHSB; 1/100), Rabbit anti-Caspase3 (Cell Signaling Technology, 1/300), rabbit anti-phospho-Histone H3 (Millipore; 1/400), and mouse anti-Islet1 (DSHB; 1/100), rabbit anti-Tbx1 (Isbio Ls-C31179, 1/100), goat anti-Tbx5 (Santa Cruz sc-7866, 1/250).

## X-gal staining

X-gal staining was performed on whole-mount embryos as previously described (*Roux et al., 2015*). For each experiment, a minimum of three embryos of each genotype was observed. Embryos were examined using an AxioZoom.V16 microscope (Zeiss) and photographed with an Axiocam digital camera (Zen 2011, Zeiss).

## In situ hybridization

Whole-mount in situ hybridization (WISH) was performed as previously described (*Roux et al., 2015*). The riboprobe used in this study was *Mlc2v*. For WISH, hybridization signals were detected by alkaline phosphatase (AP)-conjugated anti-DIG antibodies (1/2000; Roche), followed by color development with NBT/BCIP (magenta) substrate (Promega). After staining, the samples were washed in PBS and post-fixed.

RNA-FISH was performed according to the protocol of the RNAscope Multiplex Fluorescent v2 Assay (Acdbio; cat. no.323110), which detects single mRNA molecules. In briefly, E8.5 and E9.5 embryos were fixed for 20–30 hr in 4% paraformaldehyde and then dehydrated in methanol. Whole-mount RNA-FISH was performed as previously described (*de Soysa et al., 2019*). Embryos were imaged using an AxioZoom.V16 microscope (Zeiss). The following probes were used: mm-*Hoxb1*-C1 (Acdbio; cat no. 541861), mm-*Nppa* (cat no. 418691), mm-*Bmp4*-C1 (cat no. 401301), mm-*Gata4* (cat no. 417881), mm-*Aldh1a2* (cat no. 447391), mm-e*GFP*-C3 (cat no. 400281), mm-*Isl1* (cat no. 451931), mm-*Nr2f2*-C3 (cat no. 480301), mm-*tdTomato*-C3 (cat no. 317041), mm-*Tbx1*-C1 (cat no. 481911), mm-*Tbx5*-C2 (cat no. 519581), and mm-*Osr1*-C2 (cat no. 496281).

## ATAC-seq

For each sample, 10,000 FACS-sorted cells were used (n > 3 embryos for each genotype). Cell preparation, transposition reaction, and library amplification were performed as previously described (*Buenrostro et al., 2013*). Paired-end deep sequencing was performed using a service from GenomEast platform (IGBMC, Strasbourg).

Processing of ATAC-sequencing data and statistical analysis. Raw ATAC-Seq reads were aligned with the SNAP aligner (http://snap.cs.berkeley.edu/) on the reference GRCm38 mouse genome. Deduplicated reads were marked and, following ENCODE specifications (https://www.encodeproject.org/atac-seq/), unmapped, not primarily aligned, failing platform and duplicated reads were removed using samtools (-F 1804). Properly paired reads were kept (samtools -f 2). Finally, reads mapping blacklisted regions for mouse genome mm10 provided by ENCODE (*Carroll et al., 2014*) were excluded from the analysis.

To evaluate reproducibility between replicates and retain peaks with high rank consistency, we applied the Irreproducible Discovery Rate (IDR; https://f1000.com/work) methodology from ENCODE. Only peaks with an IDR value lower than 0.1 were retained.

Narrow peaks were called with MACS 2.1.1. (https://f1000.com/work) BigWig files were generated from bedGraph files to visualize fold enrichment and *p*-value for all regions within UCSC genome browser.

A Volcano Plot was used to represent peaks in function of the $\log_2$ ratio between two conditions, and the adjusted *p*-value scaled as -log10. A peak located on the upper right part of the plot corresponds to a significantly strongly enriched peak in the GFP+ (*Hoxb1-GFP*) population.

A MA plot ($\log_2$ fold change vs. mean average) was used to visualize changes in chromatin accessibility for all peaks. MA plot depicts the differences between ATAC-seq peaks in the experimental samples by transforming the data onto M (log ratio) and A (mean average) scales, then plotting

these values. Differential chromatin accessibility is expressed as a log fold change of at least two folds and a *p*-value of <0.1 and reveals the relative gain of chromatin regions in GFP+ cells (above the 0 threshold line) as compared to the gain in the Tomato+ cells (below the 0 threshold line).

Differential peaks between GFP+ (*Hoxb1-GFP*) and Tomato+ (*Mef2c-Cre;Rosa$^{tdT}$*) samples were identified using a bed file containing selected peaks from IDR methodology and the DiffBind R package (10.18129/B9.bioc.DiffBind). Peaks with FDR (False Discovery Rate) at 10% were kept.

Differential peaks were annotated using the HOMER[7] software and genes in the vicinity of peaks (+ / - 150 kb from summit) were selected.

In order to perform motif analysis, we generated a Fasta file containing all sequences surrounding peak summits (+ / - 200 bp) and used the HOMER findMotifsGenome feature. Known and de novo motifs were identified. All possible order-3 combinations motifs were generated for each known and de novo motifs. To assess enrichment of motifs combinations, *p*-values were computed using 20,000 random sequences of 400 bp from the mouse genome. Fisher's exact test was applied to compare random and peaks sequences.

## RNA-seq

Total RNA was isolated from the pharyngeal region and sorted cells with NucleoSpin RNA XS (Macherey-Nagel) following the protocol of the manufacturer. cDNA was generated and amplified with the Ovation RNAseq v2 kit (NuGEN). Briefly, 2 ng of total RNA were used for mixed random-/polyA-primed first-strand cDNA synthesis. After second strand synthesis, the double-stranded cDNA was amplified by single primer isothermal amplification, and the amplified cDNA was bead-purified (AmpureXP, Beckman-Coulter). Paired-end deep-transcriptome sequencing was performed using a service from GenomEast platform (IGBMC, Strasbourg).

E9.5 *Mef2c-Cre:Rosa$^{tdTomato}$* and *Hoxb1$^{GFP}$* embryos were dissected on ice-cold PBS to isolate the SHF and cells were FACS. After FACS Tomato+ and Tomato- as well as GFP+ and GFP- cells were homogenized in Trizol (Invitrogen) using a Tissue-lyzer (Qiagen). RNA was isolated from pharyngeal region of E8.5 wild-type and *Hoxb1$^{-/-}$*, and E9.5 control and *Hoxb1$^{GoF}$;Mef2c-Cre* embryos (n = 3 embryos per each genotype). RNAs of each genotype were pooled to obtain one replicate. RNA was prepared using the standard Illumina TrueSeq RNASeq library preparation kit. Libraries were sequenced in a Hiseq Illumina sequencer using a 50 bp single end elongation protocol. For details of analyses of RNA-seq data see *Supplementary file 1*. Resulting reads were aligned and gene expression quantified using RSEM v1.2.3 (*Li and Dewey, 2011*) over mouse reference GRCm38 and Ensembl genebuild 70. Gene differential expression was analyzed using EdgeR R package (*McCarthy et al., 2012*; *Robinson et al., 2010*). Genes showing altered expression with adjusted p<0.05 were considered differentially expressed. For the set of differentially expressed genes a functional analysis was performed using Ingenuity Pathway Analysis Software and DAVID (*Huang et al., 2009*), and some of the enriched processes were selected according to relevant criteria related to the biological process studied.

## RNA-Seq data processing

Raw RNA-seq reads were aligned using the STAR aligner version 2.5.2 (*Dobin et al., 2013*) on the reference GRCm38 mouse genome. Coverage visualization files (WIG) were generated with the STAR aligner software and were converted into BigWig files using UCSC wigToBigWig files to allow their visualization within the UCSC genome Browser.

In parallel, transcripts abundance was computed using HTSEQ-count 0.9.1 (*Anders et al., 2015*) and the background was estimated through read counts from intergenic regions using windows of 5 kb length.

Normalization and differential gene expression analysis between conditions were performed using R (R version 3.3.4) and DESEQ2 (*Love et al., 2014*).

For each sample, genes with null expression were removed and we set the 95th percentile of the intergenic read counts as the threshold of detection (log$_2$(normalized count + 1)). Heatmap were generated with the Pheatmap R package.

## Functional annotation

For RNA-seq and ATAC-seq, genes lists were annotated with the ClusterProfiler (*Yu et al., 2012*) system. Circos plots were generated with Goplot package (*Walter et al., 2015*).

## In vitro reporter assays

Luciferase reporter constructs were co-transfected with expression constructs for human *HOXB1*, *GATA4* (*Stefanovic et al., 2014*; *Singh et al., 2009*) and *NKX2-5* (*Singh et al., 2009*). Cos-7 (ATCC CRL-1651) and NIH3T3 (ATCC CRL-1658) cell lines were culture in DMEM high glucose (4.5 g/L, D-glucose, 4 nM L-Glutamine, 1 mM sodium pyruvate, 10% fetal calf serum). Plasmid transfection was performed using PEI (Polyethylenimine) transfection reagent. 24 hr after transfection, cell extracts and luciferase assays were performed following the protocol of the manufacturer (Promega). Mean luciferase activities and standard deviations were plotted as fold activation when compared with the empty expression plasmid. Cos-7 and NIH3T3 are a fibroblast-like cell lines suitable for transfection. Their mycoplasma contamination status resulted negative.

## Quantitative RT-PCR analysis

Total RNA was isolated from pharyngeal regions and sorted cells with NucleoSpin RNA XS (Macherey-Nagel) following the protocol of the manufacturer. cDNA was generated using the AffinityScript Multiple Temperature cDNA synthesis kit (Agilent). The expression level of different genes was assessed with quantitative real-time PCR. LightCycler 480 SYBR Green I Master mix (Thermo Fisher Scientific) was used for quantitative real-time qRT-PCR analysis with a LightCycler 480 (Roche Diagnostics) following the manufacturer's instructions. All primers are described in the Key Resources Table. Values were normalized to HPRT expression levels.

## Statistics

Statistical analyses were performed using unpaired two-tailed t-test to assess differences between two groups. Data are presented as mean ± SD. A *p*-value of <0.05 was considered significant.

## Acknowledgements

We are grateful to Heather Etchevers for her comments on the manuscript. We thank the animal facility, particularly Adeline Gatha, and Françoise Mallet for the cell sorting (INSERM U1068/CNRS UMR 7258). Sequencing was performed by the IGBMC Microarray and Sequencing platform. This work was supported by the FG National Infrastructure, funded as part of the 'Investissements d'Avenir' program managed by the Agence Nationale pour la Recherche (ANR-10-INBS-0009). We thank the Leducq Foundation for funding the program 'Imaging cardiac cell lineages at the origin of congenital heart disease' in the frame of the Research Equipment and Technological Platform Awards (Coordinator MP and Partners RK, SZ) and the Transaltantic Network of Excellence 15CVD01 (RK). This work was supported by the INSERM, the Agence National pour la Recherche (Transcardiac to SZ, MP and RK and Heartbox projects to SZ and RK), the Association Française contre les Myopathies (SZ), and the Fondation pour la Recherche Medicale DEQ20150331717 (RK). FL was supported by post-doctoral awards from l'Institut de France Lefoulon-Delalande. SS was supported by post-doctoral awards from l'Institut de France Lefoulon-Delalande and H2020-MSCA-IF-2014. BL was supported by a post-doctoral fellowship from the Fondation pour la Recherche Médicale.

## Additional information

### Funding

| Funder | Grant reference number | Author |
| --- | --- | --- |
| Agence Nationale de la Recherche | ANR-13-BSV2-0003 | Michel Puceat<br>Robert G Kelly<br>Stephane Zaffran |
| Agence Nationale de la Recherche | ANR-18-CE13-0011 | Robert G Kelly<br>Stephane Zaffran |

| | | |
|---|---|---|
| Fondation Lefoulon Delalande | | Sonia Stefanovic<br>Fabienne Lescroart |
| Association Française contre les Myopathies | MNH-Decrypt | Stephane Zaffran |
| Fondation pour la Recherche Médicale | | Brigitte Laforest |
| Fondation pour la Recherche Médicale | DEQ20150331717 | Robert G Kelly |
| European Commission | H2020-MSCA-IF-2014 | Sonia Stefanovic |
| Fondation Leducq | Research Equipment and Technological Platform Awards | Michel Puceat<br>Robert G Kelly<br>Stephane Zaffran |

The funders had no role in study design, data collection and interpretation, or the decision to submit the work for publication.

## Author contributions

Sonia Stefanovic, Conceptualization, Formal analysis, Validation, Investigation, Writing - original draft, Writing - review and editing; Brigitte Laforest, Conceptualization, Formal analysis, Validation, Investigation, Writing - review and editing; Jean-Pierre Desvignes, Conceptualization, Formal analysis, Methodology; Fabienne Lescroart, Conceptualization, Formal analysis, Validation, Investigation; Laurent Argiro, Formal analysis, Validation, Investigation; Corinne Maurel-Zaffran, Formal analysis, Investigation; David Salgado, Conceptualization, Formal analysis; Elise Plaindoux, Data curation, Formal analysis; Christopher De Bono, Data curation; Kristijan Pazur, Magali Théveniau-Ruissy, Christophe Béroud, Michel Puceat, Formal analysis; Anthony Gavalas, Resources; Robert G Kelly, Resources, Funding acquisition, Writing - review and editing; Stephane Zaffran, Conceptualization, Resources, Data curation, Supervision, Funding acquisition, Methodology, Writing - original draft, Project administration, Writing - review and editing

## Author ORCIDs

Brigitte Laforest (iD) https://orcid.org/0000-0001-6919-8922
Fabienne Lescroart (iD) http://orcid.org/0000-0003-4942-7921
Magali Théveniau-Ruissy (iD) http://orcid.org/0000-0002-7346-7096
Michel Puceat (iD) http://orcid.org/0000-0001-9055-7563
Stephane Zaffran (iD) https://orcid.org/0000-0002-0811-418X

## Ethics

Animal experimentation: This study was performed in strict accordance with the recommendations in the Guide for the Care and Use of Laboratory Animals of the National Institutes of Health. All animal procedures were carried out under protocols approved by a national appointed ethical committee for animal experimentation (Ministère de l'Education Nationale, de l'Enseignement Supérieur et de la Recherche; Authorization N°32-08102012).

## Decision letter and Author response

Decision letter https://doi.org/10.7554/eLife.55124.sa1
Author response https://doi.org/10.7554/eLife.55124.sa2

# Additional files

## Supplementary files

• Supplementary file 1. Excel file containing the list of deregulated genes identified by RNA-seq analysis (sheet one, upregulated genes; sheet two, downregulated genes).

• Supplementary file 2. Excel file containing the list of regions of open chromatin identified by ATAC-seq analysis (sheet one, upregulated genes; sheet two, downregulated genes).

• Transparent reporting form

### Data availability

Sequencing data have been deposited in Gene Expression Omnibus (https://www.ncbi.nlm.nih.gov/geo/) under accession number GSE123765 (ATAC-seq on GFP+ and Tomato+ cells); GSE123771 (RNA-seq on GFP+ and Tomato+ cells); GSE123772 (RNA-seq on Hoxb1GoF vs. control embryos) and GSE123773 (RNA-seq on Hoxb 1-/- vs. wild-type embryos). Further data has been included in the supporting files and source data files have been provided for Figures 2 and 3.

The following datasets were generated:

| Author(s) | Year | Dataset title | Dataset URL | Database and Identifier |
|---|---|---|---|---|
| Stefanovic S, Desvignes JP, Zaffran S | 2020 | Subpopulations second heart field ATAC-seq | https://www.ncbi.nlm.nih.gov/geo/query/acc.cgi?acc=GSE123765 | NCBI Gene Expression Omnibus, GSE123765 |
| Stefanovic S, Desvignes JP, Zaffran S | 2020 | Subpopulations second heart field RNA-seq | https://www.ncbi.nlm.nih.gov/geo/query/acc.cgi?acc=GSE123771 | NCBI Gene Expression Omnibus, GSE123771 |
| Stefanovic S, Desvignes JP, Zaffran S | 2020 | Hoxb1 LoF RNA-seq | https://www.ncbi.nlm.nih.gov/geo/query/acc.cgi?acc=GSE123773 | NCBI Gene Expression Omnibus, GSE123773 |

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
