## [Decision Letter]

**Acceptance summary:**

The authors have considerably strengthened the manuscript by providing ATACseq data on dissected progenitor regions of Hoxb1-aSHF over-expressing embryos, and addressed virtually all of the other issues highlighted by the reviewers. This is a detailed and nuanced study that advances our understanding of regulation of the anterior and posterior domains of the SHF, and sets up additional key questions about the role of Hoxa1 and b1, and retinoic acid signaling, in the posterior domain. The study also establishes a connection between Hoxa1/b1 function and atrial septation. The issues addressed in this paper are key to understanding heart developmental patterning more broadly, and the mechanism of congenital heart disease.

**Decision letter after peer review:**

Thank you for submitting your article "Hox-dependent coordination of cardiac progenitor cell patterning and differentiation" for consideration by *eLife*. Your article has been reviewed by three peer reviewers, one of whom is a member of our Board of Reviewing Editors, and the evaluation has been overseen by Didier Stainier as the Senior Editor. The reviewers have opted to remain anonymous.

The reviewers have discussed the reviews with one another and the Reviewing Editor has drafted this decision to help you prepare a revised submission.

Summary:

This paper by Stefanovic and Zaffran and colleagues details the role of Hoxa1 and b1 in second heart field (SHF) development. The team has used genetic models, fluorescent tags specific for anterior and posterior SHF enabling FACS isolation of minor anterior and posterior field populations, ATACseq and ChIP to explore chromatin openness and transcription factor binding, with some attempt at validation of concepts in a mouse ES cell system. The paper provides descriptive data about differentially expressed genes in the two populations, and chromatin accessibility in the two populations of cardiac progenitor cells. Among the ATAC-seq peaks that were identified to have increased accessibility in pSHF was a peak around the promoter of the Hoxb1 gene. Moreover, within differentially-accessible peaks, the Hox binding site was identified as the most over-represented potential cis-acting element. Therefore, the role of Hox genes in regulation of the a/pSHF patterning became the focus of the paper. To gain additional insight into SHF development, the authors generated a conditional GOF Hoxb1 allele. Mis-expression of Hoxb1 in the Mef2c-cre domain (primarily aSHF but also a subset of pSHF) led to disrupted RV development due to increased apoptosis of aSHF progenitor cells. RNA-seq identified many mis-regulated genes. Consistent with the disrupted differentiation in Hoxb1 GOF mice, the authors found that differentiation of aSHF progenitors was premature in Hoxb1 LOF mice. Re-analysis of Hoxb1 and a1 mutant mice confirm the presence of defects in structures related to SHF deployment, and in a/pSHF boundary. To try to provide some mechanistic insights, the authors also perform cell culture-based studies with an Nppa promoter-reporter construct. Nkx2-5 alone or Nkx2-5 + Gata4 co-transfected with the Nppa reporter in Cos-7 cells resulted in activation, as previously reported. Addition of a Hoxb1 expression plasmid repressed that activation. From this, the authors conclude that Hoxb1 may represses pSHF differentiation by directly repressing myocardial differentiation genes.

The paper articulates well the complex issues and problems surrounding SHF development. It presents the first analysis of ATACseq data relevant to the SHF and together with region-specific RNAseq data, the work presents a valuable resource. The overarching findings are that Hoxa1/b1 act redundantly to specify an important lineage boundary in the SHF delineating anterior and posterior compartments, and to regulate the timing and spatiotemporal aspects of SHF CM progenitor differentiation. The Hox genes may do this by interacting with SHF pathways, potentially involving interaction between Tbx5 posteriorly and Tbx1 anteriorly (De Bono et al), such that defects in Hoxb1/a1 expression alter the boundary between anterior and posterior compartments. These Hox genes appear to repress/delay the differentiation of pSHF cells, and in doing so are important for specification of cardiac structures including the dorsal mesenchymal protrusion (DMP) critical of formation the primary atrial setum and integration of the tissues of the central atrio-ventricular septal complex.

Essential revisions:

Although there is considerable merit in this paper and a significant amount of genome-wide profiling data presented, these data are generally descriptive. In the end, the mechanistic understanding of Hox gene function in SHF patterning is fairly weak. The general importance of Hoxb1 in SHF development is already largely known, as in their previous work (Roux et al., 2015) it was shown that loss of Hoxb1 resulted in reduced proliferation, premature differentiation, and subsequent ventricular septal defects. The re-analysis of LOF mutants presented in this paper provides greater focus on the a/pSHF boundary issues at hand; however, while the demonstration of a slight shift in the a/pSHF boundary and loss of the DMP in double mutants is important, the advance if fairly incremental and the new LOF analysis is not that novel. Much is made of the GOF model; however, this must be interpreted cautiously given the likely very high level of over-expression of Hoxb1 in the aSHF. Indeed, it is possible that the RV hypoplasia shown in the GOF model, driven by increased apoptosis in SHF cells, could be the result of over-expression. The over-expression of Hoxb1 in ES cells in which Hoxb1 levels reach 2x that of normal (no dox; Figure 7C) appears to confirms some aspects of the GOF phenotype (suppression of differentiation and changes in BMP1, TBX5 and OSR1 expression) and this is important for the model, but cell death was not changed. Moreover, in the in vitro assay , the data provided is on a single promoter construct analyzed in cell culture-based studies in Cos-7 cells. These experiments are helpful but limited in scope and context, and do not provide compelling mechanistic support or nuanced information for the model.

While on balanced the reviewers are supportive of the direction of this study, they feel that more specific support for the role of Hoxb1 in SHF development will be essential for publication. This could involve asking is Hox GoF or LoF alter chromatin accessibility of key differentiation and patterning genes, or provision of appropriate studies extending our understanding of Hox function in SHF development using in vivo or more contextual assays. We are prepared to extend the revision period to 4 months if the authors think they can provide additional support for the proposed model. The authors should also address the individual points made by the reviewers.

---

## [Author Response]

Essential revisions:Although there is considerable merit in this paper and a significant amount of genome-wide profiling data presented, these data are generally descriptive. In the end, the mechanistic understanding of Hox gene function in SHF patterning is fairly weak. The general importance of Hoxb1 in SHF development is already largely known, as in their previous work (Roux et al., 2015) it was shown that loss of Hoxb1 resulted in reduced proliferation, premature differentiation, and subsequent ventricular septal defects. The re-analysis of LOF mutants presented in this paper provides greater focus on the a/pSHF boundary issues at hand; however, while the demonstration of a slight shift in the a/pSHF boundary and loss of the DMP in double mutants is important, the advance if fairly incremental and the new LOF analysis is not that novel. Much is made of the GOF model; however, this must be interpreted cautiously given the likely very high level of over-expression of Hoxb1 in the aSHF. Indeed, it is possible that the RV hypoplasia shown in the GOF model, driven by increased apoptosis in SHF cells, could be the result of over-expression. The over-expression of Hoxb1 in ES cells in which Hoxb1 levels reach 2x that of normal (no dox; Figure 7C) appears to confirms some aspects of the GOF phenotype (suppression of differentiation and changes in BMP1, TBX5 and OSR1 expression) and this is important for the model, but cell death was not changed. Moreover, in the in vitro assay , the data provided is on a single promoter construct analyzed in cell culture-based studies in Cos-7 cells. These experiments are helpful but limited in scope and context, and do not provide compelling mechanistic support or nuanced information for the model.

We have extended our analysis to the already characterized *Tnnt2* promoter and provided evidence that Hoxb1 also represses this myocardial gene. These data are now shown in the revised manuscript (subsection “Hoxb1 represses the expression of the differentiation marker *Nppa*”, Figure 8).

While on balanced the reviewers are supportive of the direction of this study, they feel that more specific support for the role of Hoxb1 in SHF development will be essential for publication. This could involve asking is Hox GoF or LoF alter chromatin accessibility of key differentiation and patterning genes, or provision of appropriate studies extending our understanding of Hox function in SHF development using in vivo or more contextual assays.

To further examine whether ectopic expression of Hoxb1 GoF can alter chromatin accessibility we have performed new ATAC-seq experiments on aSHF regions microdissected from Hoxb1GoF;Mef2c-Cre and Mef2c-Cre;RosaTomato (control) embryos. We have now provided this novel data in the revised manuscript (Figure 5—figure supplement 7). Comparative analysis of ATAC-seq data showed that Hoxb1GoF alters the chromatin accessibility of many genes.

A major criticism of our manuscript concerns the level of over-expression of Hoxb1 in the aSHF. “Indeed, it is possible that the RV hypoplasia shown in the GOF model, driven by increased apoptosis in SHF cells, could be the result of over-expression“.

We agree that this point is very important. We have now provided data showing that Hoxb1mRNAs levels increased 2-fold in Hoxb1 GoF embryos compared to controls. In addition, we have used Hoxb1-Cre to over-express Hoxb1 in the pSHF and showed that this over-expression did not affect the structure of the heart, suggesting that the over-expression does not activate apoptosis in SHF cells and demonstrating that the phenotype results from misexpression of Hoxb1 rather than overexpression(Figure 4—figure supplement 5).